# Reinforcement learning establishes a minimal metacognitive process to monitor and control motor learning performance

Taisei Sugiyama [1], Nicolas Schweighofer[2] & Jun Izawa [3] ✉

Humans and animals develop learning-to-learn strategies throughout their lives to accelerate learning. One theory suggests that this is achieved by a metacognitive process of controlling and monitoring learning. Although such learning-to-learn is also observed in motor learning, the metacognitive aspect of learning regulation has not been considered in classical theories of motor learning. Here, we formulated a minimal mechanism of this process as reinforcement learning of motor learning properties, which regulates a policy for memory update in response to sensory prediction error while monitoring its performance. This theory was confirmed in human motor learning experiments, in which the subjective sense of learning-outcome association determined the direction of up- and down-regulation of both learning speed and memory retention. Thus, it provides a simple, unifying account for variations in learning speeds, where the reinforcement learning mechanism monitors and controls the motor learning process.

Learning ability is not fixed but improves through life-long training. A theory in educational psychology suggests that learning-to-learn is achieved by metacognitive strategies of monitoring and controlling learning processes, which develop gradually and are automated over a long period[1–3]. This theory has been applied to understand explicit learning ability, such as the students' academic performance[1–3]. On the other hand, little is known about whether humans can also monitor and control implicit learning processes, such as motor learning. Because such recognition of automatic implicit process would appear to contradict the well-accepted theory of implicit motor adaptation[4–6], the metacognitive aspect of motor learning remains unexplored. Nevertheless, humans show flexible learning behavior in motor learning[7–12], which can be viewed as a manifestation of learning-to-learn, or "meta-learning". How can we monitor and control implicit motor learning?

Several factors are known to change properties of motor learning, such as learning speed and memory retention[8,11,12]. For example, learning accelerates when learners repeatedly experience the same environments (i.e., low volatility), whereas it decelerates when they

repeatedly experience rapidly changing environments (i.e., high volatility)[8]. In addition, motivational signals also influence motor learning[11,12]. For instance, rewards and punishments influence learning speeds, although their effects are variable; some studies show an increase in learning speed[12] while others show no effects[11]. Since these various factors are seemingly unrelated to each other, different theories have been proposed to explain these phenomena through different mechanisms[8,10,12–15]. However, in fact, there is a common characteristic behind these various factors. Namely, they influence the outcome of motor learning, in principle, regardless of whether it affects the outcome directly (e.g., by adding monetary rewards to task performance)[11,12] or indirectly (e.g., by destabilizing environments to make learning ineffective)[8,10]. Thus, if humans value or devalue motor learning according to these outcomes, such subjective evaluation could promote or suppress motor learning.

Here, we sought a minimal framework for this possible metacognitive process for learning-to-learn, also called meta-learning, by theorizing it based on the reinforcement learning of implicit motor learning. In our theory, the update of the motor learning properties

[1]Empowerment Informatics, University of Tsukuba, Tsukuba, Ibaraki 305-8573, Japan. [2]Biokinesiology and Physical Therapy, University of Southern California, Los Angeles, CA 90089-9006, USA. [3]Institute of Systems and Information Engineering, University of Tsukuba, Tsukuba, Ibaraki 305-8573, Japan. ✉e-mail: izawa@emp.tsukuba.ac.jp

follows the learning-outcome structure according to rewards and punishments. We then derived an experimental paradigm based on this theory to examine the flexibility of meta-learning ability for motor learning.

## Results

### Theory of meta-learning as reinforcement learning of a memory update policy

Instead of assuming a combination of multiple learning rates[14,16] or a varying learning rate[8,13], as proposed in previous motor learning studies, we begin with a simple assumption that the motor memory $x^{(k)}$ on trial $k$ is updated by a "memory update action" $u^{(k)}$ over trials, $x^{(k+1)} = x^{(k)} + u^{(k)}$. In this framework, motor learning is a sequential decision-making process in the state space spanned by memory $x^{(k)}$ and sensory prediction error $e^{(k)}$ where the action $u^{(k)}$ determines the next memory $x^{(k+1)}$ (Fig. 1A). The memory update action $u^{(k)}$ is drawn from a policy function, which is characterized by a meta-

parameter $\theta$ composed of a learning rate $\beta$ that characterizes the speed of learning and the retention rate $\alpha$ that characterizes the speed of forgetting, consistent with previous motor learning theories[17], where the memory update is described by $x^{(k+1)} = \alpha x^{(k)} + \beta e^{(k)}$. Assuming that the ultimate goal of learning is to maximize rewards and minimize punishments, instead of simply minimizing errors, we derive an update rule of the meta-parameters of motor learning $(\alpha, \beta)$ by the policy gradient theorem[18] (see methods for derivation): $\alpha \leftarrow \alpha + \frac{\eta_\alpha}{\sigma_x^2} \cdot n_x^{(k)} \cdot r^{(k+1)} \cdot x^{(k)}$ and $\beta \leftarrow \beta + \frac{\eta_\beta}{\sigma_x^2} \cdot n_x^{(k)} \cdot r^{(k+1)} \cdot e^{(k)}$, where $\eta_\alpha, \eta_\beta$ are meta-learning rates, $\sigma_x^2$ is the memory noise variance, $n_x^{(k)}$ is the memory noise, $r^{(k+1)}$ is the outcome of memory update. For example, when the memory update yields a reward outcome $r(x^{(k+1)})$, the meta-parameters $\alpha$ and $\beta$ are updated in the direction of $n_x^{(k)}$ scaled by the evaluation of the current memory update $r(x^{(k+1)})$ and the contributions of the current $\alpha$ and $\beta$ to the memory update ($x^{(k)}$ and $e^{(k)}$, respectively). Here, memory noise $n_x^{(k)}$ acts as exploration noise. Thus, reinforcement learning establishes meta-learning by integrating the

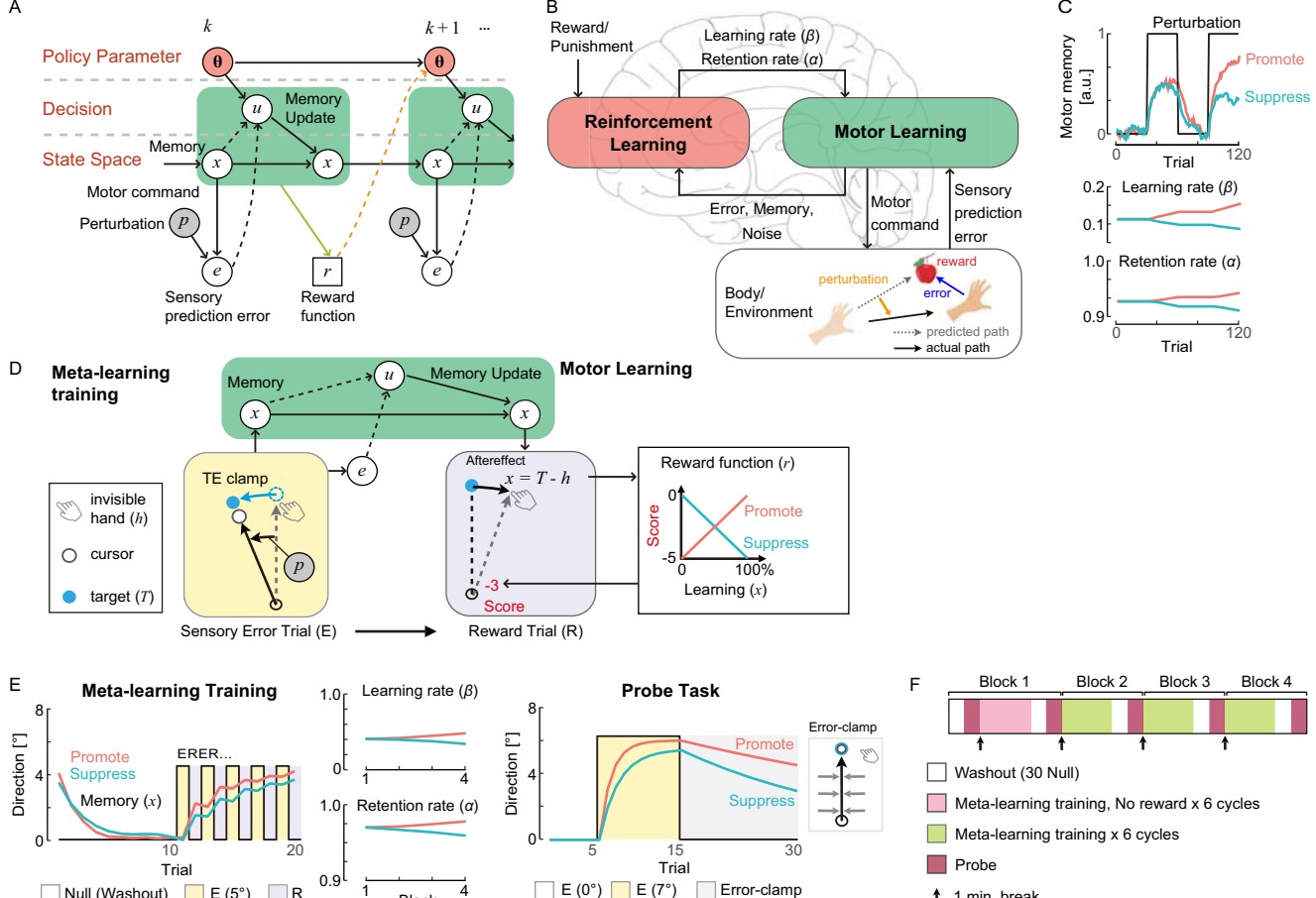

**Fig. 1 | Meta-learning theory and paradigm. A** Motor learning as a sequential decision-making process. The action $u^{(k)}$ updates the memory and sensory prediction error states $\{x^{(k)}, e^{(k)}\}$ to the next states $\{x^{(k+1)}, e^{(k+1)}\}$, and generates a reward $r(x^{(x+1)})$ in the given environment ($p$: perturbation). The action $u^{(k)}$ responds to $\{x^{(k)}, e^{(k)}\}$, characterized by meta-parameter $\theta$ and influenced by memory noise $n_x^{(k)}$, i.e., drawn from a policy distribution $u^{(k)} \sim \pi_\theta(u^{(k)}|x^{(k)}, e^{(k)})$[42]. This aligns with the previous models of error-based motor learning $x^{(k+1)} = \alpha x^{(k)} + \beta e^{(k)} + n_x^{(k)}$ ($\alpha$: retention rate, $\beta$: learning rate)[17,21,23] when the learner has a linear policy function. **B** The primary hypothesis of this study is that the meta-parameter $\theta = [\alpha, \beta]^T$ is updated by reinforcement learning rule (policy gradient[18]) to maximize rewards and minimize punishments: $\theta \leftarrow \theta + \nabla_\theta \log \pi_\theta \cdot r(x^{(x+1)})$. **C** Simulated change of meta-parameters in the two opposite reward functions. Reward is given for learning in "**Promote**" (magenta) and for not-learning in "**Suppress**" (cyan). Reinforcement learning upregulates $\theta = [\alpha, \beta]^T$ to learn faster in **Promote**, whereas it downregulates them to

learn slower in **Suppress**. a.u. = arbitrary unit. **D** Meta-learning training. Learners experience a sensory-error (E) trial where the sensory prediction error $e$ is induced by cursor rotation $p$ while the task error is clamped (TE clamp). Subsequently, they experience a reward (R) trial in which the updated memory $u$, manifested as an aftereffect $h = T - x$, is evaluated with reward function $r$. **Promote** and **Suppress** were implemented by linking the aftereffect and reward oppositely. Reward is delivered as a numerical score associated with monetary reward. **E** The task schedule. Learners repeat meta-learning training that comprises pairs of E and R trials and Null trials (in which the veridical cursor feedback was given). After every six repetitions of training, they perform a probe task, developed from previously established motor learning paradigms to estimate learning parameters. The simulated reach behavior and changes in $\theta$ are plotted for **Promote** and **Suppress**. **F** The task is separated into four blocks, and behavior is analyzed block-by-block. The first block marked in pink is the baseline condition in which score is absent in R trials.

internal monitoring of the current motor learning states (memory and sensory prediction error) with the learning performance (i.e., rewards) in order to control memory updates and retentions (Fig. 1B).

We simulated this model in a typical perturbation sequence composed of baseline trials, initial perturbation trials for learning, washout trials, and subsequent perturbation trials for re-learning (Fig. 1C). We considered two reward functions with different learning-reward associations (learning-outcome structure). In one condition (Promote learning [**Promote**]), which encouraged fast motor learning, the reward function was designed to generate larger rewards following larger motor memory updates. In the other condition (Suppress learning [**Suppress**]), which encouraged slow motor learning, the reward function was designed to generate larger rewards following smaller motor memory updates. Simulation results (Fig. 1C) show that reinforcement learning updates the learning and retention rates $\alpha$ and $\beta$ in a manner congruent with the two conditions; in the **Promote** condition, both $\alpha$ and $\beta$ increase, leading to greater memory update. Contrarily, in the **Suppress** condition, both $\alpha$ and $\beta$ decrease, leading to lower memory update. Therefore, the theory predicts both the acceleration of learning[19] and the deceleration of learning[20].

## Human motor learning experiments

To examine whether humans show behavior comparable to the simulated meta-learning agent, we modified previous motor learning tasks[9,21] to a meta-learning training paradigm, in which the motor memory update induced by sensory prediction error in one trial (Sensory Error trial [E]) was evaluated by a numerical score associated with a monetary compensation in the subsequent trial (Reward trial [R]) (Fig. 1D). Crucially, this dissociation between presentation of the sensory prediction error and the reward in two adjacent trials enables us to manipulate the learning-outcome association for meta-learning independent of the learning-error association for motor learning. For instance, in a conventional motor learning task, large learning induces small errors in subsequent trials, which always encourages learning. In contrast, with this dissociation, we can encourage (**Promote**) or discourage (**Suppress**) motor learning.

Eighty healthy participants gave informed consent before participating in the experiment, which was approved by the Institutional Review Board at the University of Tsukuba. A typical apparatus for visuomotor adaptation tasks was used[22]. In brief, participants held a planar robot manipulandum and made rapid, horizontal, arm-reaching movements to hit targets displayed in front of them. A mirror above the manipulandum occluded the hand. Visual stimuli were presented on a monitor, reflected in the mirror, such that they appeared at the same height as the hand.

Participants performed meta-learning training in which pairs of Sensory Error (E) and Reward (R) trials were repeated (Figs. 1D, E, S1). In both E and R trials, the target blinked once 10 cm from the start position, and participants were asked to hit the target with their hand as accurately as possible. In E trials, unbeknownst to the participants, online cursor feedback representing the hand position was rotated by 5° from its actual position during movement. This perturbation induced a sensory prediction error and a memory update in response to this error[21–24]. In addition, to remove any task-performance feedback generated by the error between the target and the cursor (task error[9,23]), which could act as a reward/punishment signal and modulate learning, we used a task error clamp method (TE clamp) such that E trials always appeared successful to participants[9]. Specifically, when the cursor reached 8 cm, the target blinked again at 10 cm, but its angle was shifted in the direction of the cursor movement[9], unbeknownst to the participants. In R trials, no cursor feedback or second target blink was provided. Instead, following the movement, participants were shown a numerical score proportional to the motor memory updated in the E trial. This update was assessed by the aftereffect, i.e., the

angular error between the target direction and the hand direction. Participants were informed that positive or negative scores were associated with monetary gains or losses, respectively. As in the simulations described above, we compared the effects of two score functions (**Promote** and **Suppress**) that determined the relationship between the size of the memory update and the score.

**Experiment 1: Negative scores.** Since human reinforcement learning has greater sensitivity to negative reward prediction errors[25,26], we first tested the effects of negative scores in Experiment 1 ($N = 40$). The goal of reinforcement learning is, therefore, to avoid punishment. The participants' memory profiles ($x$), estimated from changes in the reach angle (see Methods), agreed with those of simulated agents in the last block of meta-learning training (Fig. 1E, Fig. 2A). Across blocks, the group difference between **Promote** and **Suppress** developed gradually, both in the initial update (first response to the error before any reward is delivered) of motor memory in the first R trial ($p = .02$) and in the accumulated update in the last R trial ($p < 10^{-5}$) (Fig. 2B). In addition, both groups increased their scores over blocks ($p < 10^{-14}$ for **Promote** and $p < 10^{-11}$ for **Suppress**, Fig. 2C). Thus, participants learned how to regulate their motor learning performance to minimize punishment, according to the learning-outcome structure (**Promote**/**Suppress**), supporting our hypothesis of meta-learning as reinforcement learning of motor learning properties.

We then assessed changes in the meta-parameters $\alpha$ and $\beta$ over blocks in Probe tasks (Fig. 1E, F). The Probe task comprised consecutive E trials, during which we examined the motor memory update in response to a 7° rotation to estimate $\beta$. These trials were followed by consecutive Error-Clamp trials, in which the error was clamped at zero by fixing the cursor movement to the target direction[27,28]. Memory decay during clamp trials allowed us to estimate $\alpha$. Note that because we continued to use the TE clamp in E trials, no reward feedback was given in Probe trials. A 7° perturbation instead of a 5° perturbation during training allowed us to verify generalization of meta-learning effects across error sizes.

The profiles over blocks of both memory updates and meta-parameters $\alpha$ and $\beta$ were qualitatively similar to those of the simulated agent in the two conditions, **Promote** and **Suppress** (compare Fig. 1E to Fig. 2D, F). Changes over blocks were significantly different between groups in both the initial learning response (the response to the cursor rotation at the 1st trial measured at the 2nd trial) ($p = 0.04$) and the average memory of all Error-Clamp trials ($p < 0.004$, Fig. 2E), consistent with changes in both the initial and the accumulated updates during meta-learning training. Although this initial learning response reflects $\beta$, it is only a point estimate of it. Similarly, although the average memory of Error-Clamp trials reflects $\alpha$, it is influenced by both the accumulated and the forgetting effects of motor memory. Thus, we then estimated the learning parameters $\alpha$ and $\beta$ in both conditions with a model-based Bayesian estimation method[29,30] using all trials in Probe (Fig. 2F; see Methods for details). Group differences were significant for both changes in $\alpha$ (95% high density interval (HDI) = [0.005, 0.020]; $p < 10^{-3}$) and $\beta$ (95%HDI = [0.007, 0.120]; $p = 0.01$). See Tables S1–S6 and S13 for a summary of these statistical analyses.

**Experiment 2: Positive scores.** Experiment 2 ($N = 40$) was identical to Experiment 1 except that negative scores were replaced with positive scores in R trials. The goal of reinforcement learning is therefore to acquire rewards. This simple change resulted in marked differences in the effects of training (Fig. 3A–C) from Experiment 1 (Fig. 2A–C). Notably, while the **Suppress** group showed an improved score ($p < 10^{-7}$), the **Promote** group did not show evidence of increasing scores during blocks ($p = 0.43$), yielding a significant group difference ($p = 0.002$) (Fig. 3C). Because of this weak meta-learning effect in the **Promote** group, no significant group differences were found either in

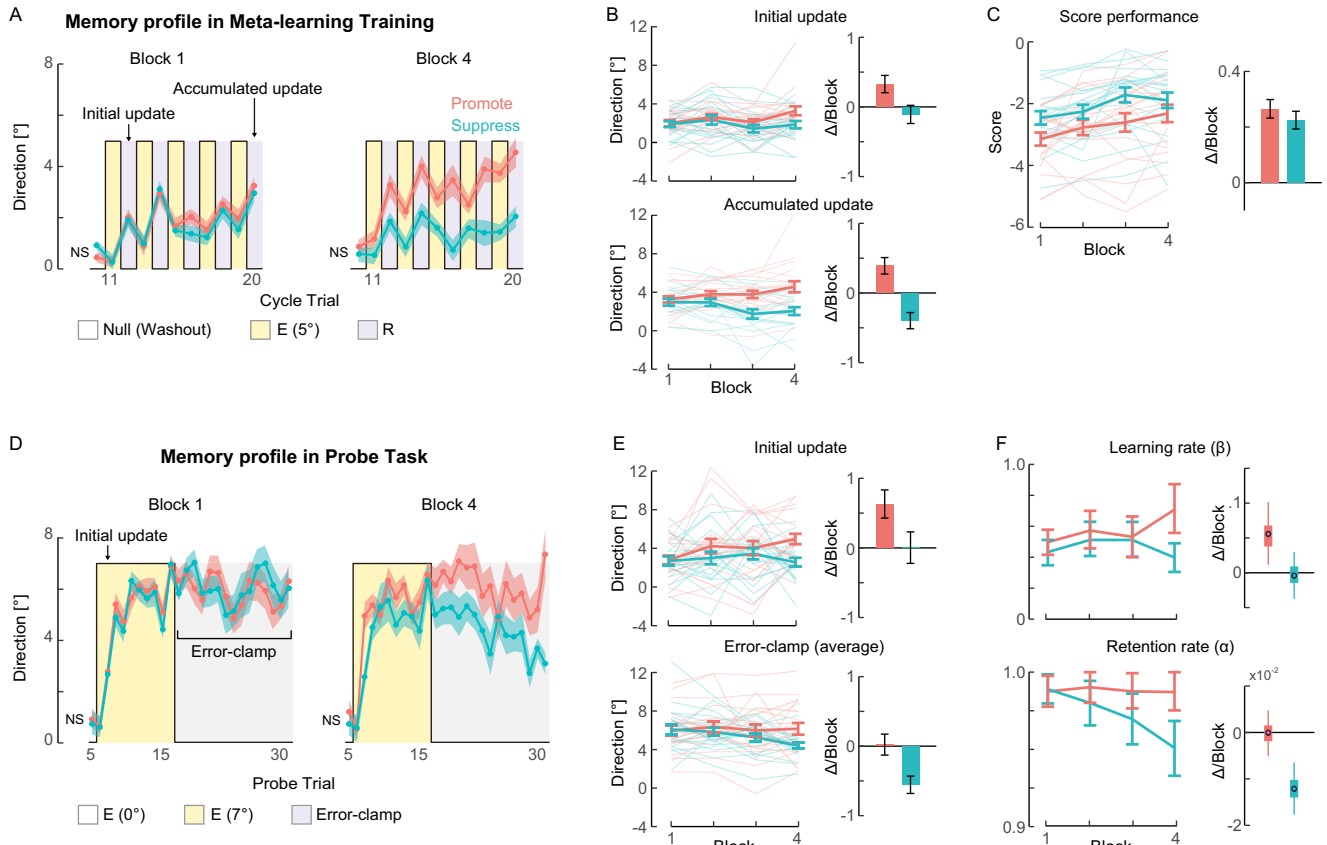

**Fig. 2 | Changes in behavior and estimated learning parameters with negative scores (Experiment 1). A** Memory profiles of the two groups in Blocks 1 and 4 of meta-learning training with opposite reward functions (**Promote**, magenta, and **Suppress**, cyan). **B** Changes during blocks and estimated linear slopes in the initial/accumulated memory updates in the first/last R trials. **C** Changes during blocks and estimated linear slopes of the average score per trial. **D** Memory profiles of the two groups in Blocks 1 and 4 of Probe. **E** Changes over blocks and estimated linear slopes in the initial memory update in the second E trial with rotation and the accumulation and retention of memory, measured as the average of all error-clamp trials. **F** Changes during blocks and estimated linear slopes

in estimated learning parameters. For all panels, thick lines/dots/circles and error bars/shading represent group means and SEMs, except the error bars in F, where box and whisker represent the 25–75th and 2.5-97.5th percentiles for the posterior density estimated by the Markov Chain Monte Carlo (MCMC) method. Faded lines represent individual participants' data in (**B**), (**C**), and (**E**). Each mean and SEM are calculated for data from 20 human participants per group (in total $N = 40$). NS indicates "Not Significant" between **Promote** and **Suppress** at the pre-perturbation baseline trial (two-sided Wilcoxon rank sum test for Training Block 1: $p = 0.95$ and Block 4: $p = 0.19$, Probe-Block 1: $p = 0.60$ and Block 4: $p = 0.90$).

initial updates ($p = 0.18$) or accumulated updates ($p = 0.10$) (Fig. 3B). In the Probe task (Fig. 3D), a significant difference was found between the groups in the average memory of Error-Clamp trials ($p = 0.01$), but not in the initial update ($p = 0.82$) (Fig. 3E). The estimated learning parameters agreed with these observations; group differences were significant in changes in $\alpha$ (95%HDI = [0.002, 0.017]; $p = 0.008$) but not in $\beta$ (95%HDI = [−0.044, 0.019]; $p = 0.21$) (Fig. 3F). See Tables S7–S12 and S14 for a summary of these statistical analyses.

**Estimation of meta-learning rates in both Experiments.** To directly examine the difference between punishment and rewards in the effect of meta-learning, the meta-learning rate ($\eta$) was estimated for each learning parameter ($\alpha$ and $\beta$) and each experiment using a Bayesian estimation method (Fig. 3G; see Methods for details). $\eta_\alpha$ was positive in both Experiment 1 (95%HDI = [$3.8 \times 10^{-6}$, $1.3 \times 10^{-4}$]; $p = 0.03$) and Experiment 2 (95%HDI = [$1.7 \times 10^{-5}$, $2.0 \times 10^{-4}$]; $p = 0.009$), with no evidence of a difference (95%HDI = [$−1.5 \times 10^{-4}$, $7.2 \times 10^{-5}$]; $p = 0.22$). On the other hand, $\eta_\beta$ was positive in Experiment 1 (95%HDI = [$3.7 \times 10^{-5}$, $5.5 \times 10^{-4}$]; $p = 0.03$), but not in Experiment 2 (95%HDI = [$−1.9 \times 10^{-4}$, $9.1 \times 10^{-5}$]; $p = 0.26$), with a significant difference (95%HDI = [$3.2 \times 10^{-5}$, $6.4 \times 10^{-4}$]; $p = 0.02$). See Table S15 for a summary of these statistical analyses, which predicted the reach angle change in Probe (Fig. S2).

## Unified explanation for previous results in motor learning research

Overall, our results demonstrate that reinforcement learning affected the properties of motor learning because participants regulated their learning and retention rates to increase their scores. The association between learning and scores (**Promote/Suppress**) determines the up- and down-regulation of both learning and retention rates, with a higher sensitivity to punishments[25,26]. We re-emphasize that both **Promote** and **Suppress** groups experienced the same sensory stimuli and perturbation sequence through both the meta-learning training and the Probe task, and that neither rewards nor task errors were provided in visuomotor rotation trials (E trials). Thus, no previous hypothesis regarding the stability of error and environment[8], uncertainty[10], reward-based learning[21], or learning context[31], can predict the change of learning and retention rates in our meta-learning training.

In contrast, our meta-learning model provides a unified theory that accounts for previous results in motor learning research. First, if we assume that task errors provide punishment information and are thus aversive[23,32–34], our model explains previous results of the change in learning speed influenced by volatility of the perturbation in the absence of apparent reward feedback[8] (Fig. 4A, B). When the direction of perturbation is constant throughout learning trials (low volatility),

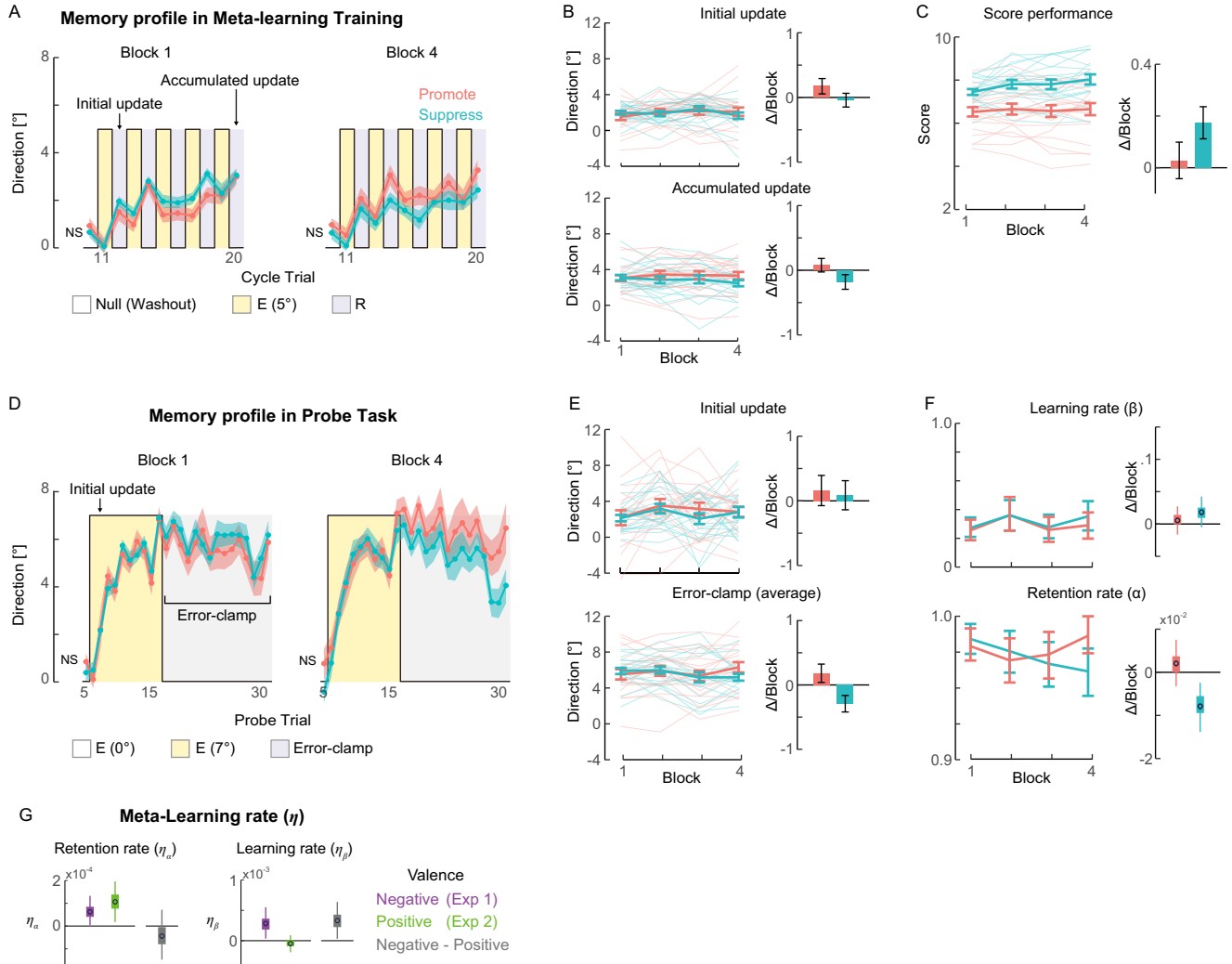

**Fig. 3 | Changes in behavior and estimated learning parameters with positive scores (Experiment 2). A** Memory profiles of the two groups in Blocks 1 and 4 of meta-learning training with opposite reward functions (**Promote**, magenta, and **Suppress**, cyan). **B** Changes during blocks and estimated linear slopes in the initial/accumulated memory updates in the first/last R trials. **C** Changes in memory size during blocks and estimated linear slopes in the average score per trial. **D** Memory (aftereffect) profiles of the two groups in Blocks 1 and 4 of Probe. **E** Changes during blocks and estimated linear slopes in the initial memory update in the second E trial with rotation and the accumulation and retention of memory, measured as the average of all error-clamp trials. **F** Changes over blocks and estimated linear slopes

in estimated learning parameters. **G** Meta-learning rates ($\eta$) for each experiment and the difference between punishments and rewards (difference between Experiments 1 and 2). For all panels, lines/dots/circles and error bars/shading represent group means and SEMs, except the error bars in (**F**) and (**G**), where box and whisker represent the 25–75th and 2.5–97.5th percentiles for the posterior density estimated by the MCMC method. Faded lines represent individual participants' data in (**B**), (**C**), and (**E**). Each mean and SEM are calculated for data from 20 human participants per group (in total $N = 40$ for **A**–**F**, $N = 80$ for **G**). NS indicates "Not Significant" (two-sided Wilcoxon rank sum test for Training Block 1: $p = 0.70$ and Block4: $p = 0.53$, Probe-Block 1: $p = 0.49$ and Block 4: $p = 0.23$).

updating motor commands in response to observed errors in a particular trial decreases errors in subsequent trials, which in turn, minimizes the punishment (the task error). Thus, reinforcement learning trains the learning policy to increase the memory update, minimizing punishment. In contrast, in volatile environments with rapidly changing perturbations, the memory update in response to prediction errors increases the error in the next trial with a different perturbation (e.g., Learning to compensate for a rightward perturbation increases the error when the perturbation moves leftward in the subsequent trial). Thus, reinforcement learning trains the learning policy to decrease the memory update, again minimizing the punishment.

Second, and for the same reason, our model also explains how manipulation of the task error modulates learning speed[9] (Fig. 4C, D). In the Standard TE (StdTE) condition[9], the learner is trained in a 30° cursor rotation, generating task error that can be reduced by motor learning, like in standard motor learning tasks. In the NoTE condition,

the task error was clamped, as with the TE clamp of our experiment (Fig. 1D), and therefore motor learning did not affect the task errors. In the RandomTE condition, the task error was randomized by randomly shifting the target from the cursor, and therefore motor learning did not systematically affect the task errors. Only StdTE shows accelerated learning speed compared to NoTE and RandomTE (Fig. 4C). Our model accounts for these results; reinforcement learning increases the learning rate to minimize punishment in StdTE, the only condition in which motor learning effectively reduces the task errors (Fig. 4D).

Third, if we further assume that the brain encodes punishment (negative score) and rewards (positive score) differently for reinforcement learning[26] such that the sign of the feedback signal biases the efficacy of reinforcement learning[25], our model also accounts for the puzzling effect of monetary feedback on motor learning, in which punishment accelerated learning while reward increased retention[11] (Fig. 4E, F). In the punishment condition[11], a negative score

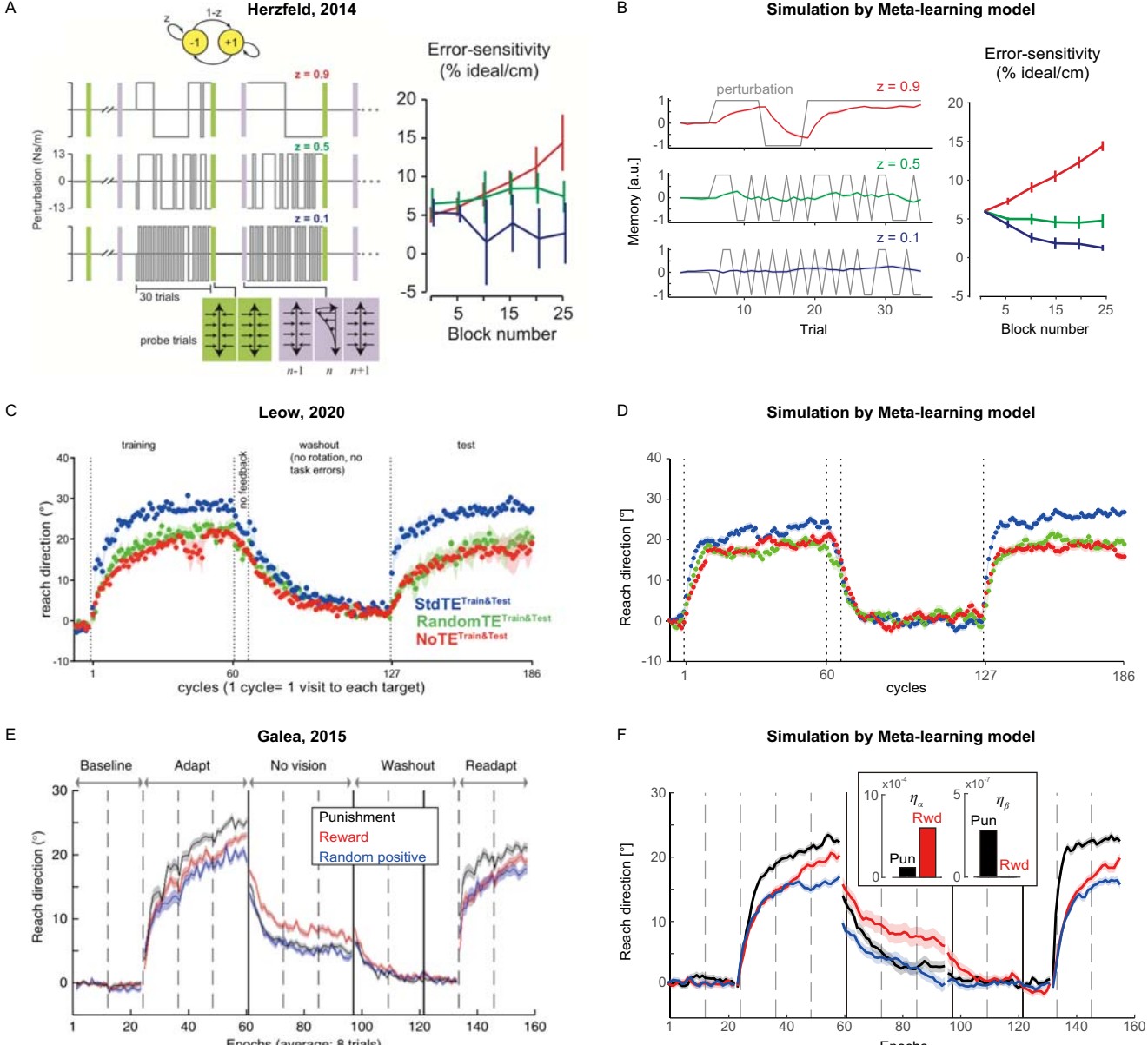

**Fig. 4 | Replication of previous reports on changes in the speed of motor learning by simulation with the present Meta-learning model.** Assuming that task error is a punishment feedback, the present meta-learning model replicated previous reports. **A**, **B** Changes in error-sensitivity (speed of learning, i.e., $\beta$) in different probabilities of flip in the perturbation direction influencing the history of error, reported in the original study (**A**) and simulated by the model (**B**). Sensitivity increased in a stable environment ($z = 0.9$, red) and decreased in a rapidly changing environment ($z = 0.1$, blue). Sensitivity did not show apparent changes in medium stability ($z = 0.5$, green). The simulated memory profile at the end of the task was also plotted for each condition (**B**, left). **C**, **D** Effects of manipulating task error on motor learning, reported in the original study (**C**) and simulated by the model (**D**). Acceleration of learning (saving, blue) disappeared when a task error was randomly given (green) or removed (red). **E**, **F** Effects of reward/punishment on trajectory error, reported in the original study (**E**) and simulated by

the model (**F**). The inset shows the used meta-learning rates for the simulation. Learning accelerated only in adaptation to punishment (black), compared to adaptation to reward (red) or random positive (blue). For all panels, lines/dots and error bars/shaded areas indicate the mean and SEM. a.u. = arbitrary unit. Panel (**A**) is from David J. Herzfeld et al., A memory of errors in sensorimotor learning. Science, 345,1349-1353 (2014). DOI:10.1126/science.1253138. Reprinted with permission from AAAS. Panel (**C**) is adapted with permission from Leow, L. A., Marinovic, W., de Rugy, A. & Carroll, T. J. Task errors drive memories that improve sensorimotor adaptation. J Neurosci. 40(15), 3075-3088 (2020). https://doi.org/10.1523/JNEUROSCI.1506-19.2020. https://creativecommons.org/licenses/by/4.0/. Panel (**E**) is reproduced with permission from Springer Nature. Galea, J., Mallia, E., Rothwell, J. et al. The dissociable effects of punishment and reward on motor learning. Nat. Neurosci. 18, 597–602 (2015). https://doi.org/10.1038/nn.3956, Springer Nature.

proportional to the size of the error was subtracted from the maximum payment. In the reward condition, a positive score negatively proportional to the size of error was added to the payment (Fig. 4E). In the random positive condition, the score was random. Our simulations replicated the different modulation effects of punishment and reward on learning speeds and retention. Interestingly, the asymmetry in the meta-learning rates $\eta_\alpha$, $\eta_\beta$ (Fig. 4F, inset, Table S16) adjusted to

replicate these previous results is similar to the asymmetry of the estimated meta-learning rates in Experiments 1 and 2 (Fig. 3G).

## Discussion

The present study demonstrates that, following the presentation of score feedback as a function of motor learning performance, human participants both increased and decreased learning rates and retention

rates to gain rewards or avoid punishments. This effect developed gradually during the training sessions, and the direction of regulation (**Promote** or **Suppress**) was determined by the learning-outcome structure. These results show that reinforcement learning is employed to regulate the learning policy. In addition, meta-learning training differed between positive and negative scores, which can have different subjective values[26]. Thus, our results demonstrate reinforcement learning mediates the metacognitive process to control motor learning by monitoring on subjective learning performance.

Machine learning scientists have recently proposed algorithms in which a high-order reinforcement learning system trains a lower-level reinforcement learning system by regulating its learning speeds[35–37]. Such a reinforcement learning view of meta-learning is sensible because reinforcement learning is a fundamental form of the learning process for humans and animals[38–41]. It is believed to enable them to optimize actions so as to maximize rewards by forming associations between selected actions and outcomes through trial-and-error[42], which mediates a broad range of cognitive functions, from pain perception[41] to moral judgement[43]. Although such a hierarchical learning system is thought to exist in the brain[44–48], direct empirical evidence of its existence is lacking. This is because, in previous work, both the meta-learning and the learning processes were driven by rewards, preventing a clear dissociation between the two[44]. In contrast, because of the separation between the update signals for the higher system (rewards) and lower systems (sensory prediction errors), our approach dissociates the higher meta-learning level from the lower learning level. Thus, we offer the empirical evidence for the existence of a reinforcement learning mechanism for learning policy in the human brain.

In contrast with our reinforcement learning view of motor learning, previous theories of motor learning postulate that the goal of motor learning is to minimize errors[17]. Thus, these theories of the regulation of learning speeds are also based on properties of errors[8,10]. Thus, each theory can account for only a subset of the dataset. However, multiple motor learning experiences of unrelated motor skills also enhance motor learning speeds[7], which is not explained by the experience of errors alone. Here, we have proposed a minimal framework of the metacognitive process in motor learning, which actively controls both learning and retention to gain rewards and avoid punishments, explaining previous reports of changes in motor learning ability by the volatility of errors[8], reward feedback on errors[11], and history of the environment[9]. In addition, our theory and the behavioral results show that, in **Suppress** condition, a learner learns not to minimize errors since less learning decreases punishment. This contradicts the hypothesis that the goal of motor learning is error minimization. In contrast, our theory suggests that the error minimization principle holds true only when motor learning is associated with maximizing rewards or minimizing punishments.

Our meta-learning theory postulates that motor learning is decision-making for memory updates. It differs from a seemingly parallel theory in motor control, which focuses on the process of selecting motor commands instead of selecting memory updates[49]. The present study proposed that motor learning is a sequential decision-making process in a space spanned by motor memory and errors, providing empirical evidence to support our theory.

Four alternative hypotheses are refuted. First, since cursor rotation (5° or 7°) and learning-outcome structure were unnoticeable to the participants, none reported the use of aiming strategies to change their reach direction[15], suggesting that they learned and adapted to the rotation without being aware of it. Note that the participants exhibit quick learning, on average showing 30–40% of learning in initial updates (Figs. 2B, D, 3B, D). However, such fast learning does not necessarily imply explicit learning because the learning rate is faster for smaller rotation[50] and a similar learning rate was reported in a recent implicit motor learning study[51]. Although the participants recognized the score feedback explicitly, it was hard for them to notice

the action-outcome association explicitly, since the presented score appeared random due to natural motor variabilities as in a typical human reinforcement learning task[52,53]. Nevertheless, the learning-outcome structure changed the initial update in response to the initial visual rotation (Fig. 2B, E). Furthermore, because no cue or task error was presented before the initial update, the participants could not infer a context to recall the memory or develop the strategy[15]. Therefore, meta-learning was not achieved by the change of the explicit strategy. Second, since this initial response to the error was generated before any reward or punishment was delivered, this meta-learning effect on the initial response was not due to the response to reward feedback. Instead, the initial update was due to the response to sensory prediction error induced by the visual rotation itself. Third, it could be argued that the reinforcement learning system recognized the 5° rotation as a cue and learned how to respond specifically to a 5° rotation. However, this was not the case because the meta-learning effect exhibited a generalization ability between two rotation sizes from 5° to 7°. Fourth, alternatively, it could also be argued that the reinforcement learning system recognized rotation of any size as a cue and learned to respond similarly for any rotation size. This was also not the case because the generalization ability was limited to the error space that had been experienced (Supplementary Note 1 and Fig. S3). Fig. S3 also shows that the error experience was generalized between and beyond 5° and 7°. Thus, the meta-learning effect was not mediated by a simple cue-response map but by modulation of the neural policy function with receptive fields over the representation of sensory prediction error[8,54,55]. This supports our hypothesis that meta-learning monitors and controls the motor learning policy function.

Our new framework offers a unified view on variable motor learning behaviors reported in conventional motor adaptation tasks affected by three different factors: volatility of environment[8], valence[11], and task error[9] (Fig. 4). To apply our theory of meta-learning to these three adaptation tasks, we hypothesized that the task error carries information of negative reinforcer, i.e., punishment. This hypothesis has not been examined in previous motor learning studies. However, a neuroimaging study reported that the task error in the reach adaptation task is correlated with the activity in the striatum[23], the main input of the basal ganglia, which is considered responsible for reinforcement learning[38,39]. In addition, it has been shown that the task error could be explicitly noticeable[5], and the noticeable error in the goal-directed behavior is aversive[34], which decreases the dopamine level in the ventral tegmental area[32,33]. Thus, this neurophysiological evidence also supports our hypothesis of the negative reinforcer contained in the task error.

In addition to such information of negative reinforcer, the task error has been shown to be a driving input to motor adaptation[5,6,13]. In one model, the task error and sensory prediction error update two different memories in parallel, whereas in another theory, these two are multiplied to update a single memory[13]. Since the task error is noticeable, it might update the explicit motor memory while the sensory prediction error updates the implicit motor memory[5]. In another model, the task error updates both implicit and explicit memories[6]. However, because these models do not assume that the punishment factor is contained in the task error, they explain only a subset of previous behaviors (Figs. S4–5, Tables S17-18). In contrast, our meta-learning theory with this punishment factor outperforms these previous models (Fig. S5D, Tables S17–18). As a note, we showed that adding the task error as a driving input of motor memory[5,6,13] does not interfere with our meta-learning model (Figs. S5E, S6). Thus, both our novel punishment factor and the established driving input factor[5,6,13] may coexist in the task error.

Our results, showing that the same punishment information leads to both promotion and suppression of the learning gain depending on the learning-outcome structure (Fig. 2), suggest an involvement of a flexible goal-directed learning mechanism for meta-learning. In humans and animals, two different neural mechanisms are involved in

the reinforcement process. One is the Pavlovian system which responds to valence in a hardwired manner; the other is the instrumental learning system which updates the action flexibly to maximize/minimize rewards/punishments for goal-directed problems[56]. The latter system has been formalized by reinforcement learning rules[57] by machine learning scientists[18,42]. Thus, our proposed computational model of meta-learning derived from reinforcement learning theory is likely mediated by the instrumental learning system where the cortico-basal ganglia network plays the central role[38–40]. This contrast with a previous report on the effect of punishment in enhancing the gain of motor learning, where the direction of the punishment effect is fixed[11]. This effect has been modeled by a hardwired system with the one-to-one relationship between the punishment and the learning gain[13]. Such a fixed Pavlovian model cannot account for our results of both promotion and suppression of the learning gain depending on the learning-outcome structure.

Finally, the present meta-learning system suggests a close interplay between reward system and sensory prediction error system in the brain. Updating motor learning policy entails the integration of reward signals with memory update signals (current memories/sensory prediction errors. See Eq. (9) in the Methods), unlike previous theories, which hold that meta-learning is driven by either of the signals (error or reward), but not both[8,48]. Because previous research suggests an involvement of the basal ganglia in reinforcement learning[38–40] and the cerebellum in sensory error-based learning[24,58], reinforcement learning on motor learning policy is likely mediated by functional connectivity between these two learning systems. Thus, anatomical projections between the basal ganglia and the cerebellum[59] could have a computational role in meta-learning. This possibility is further supported by recent reports of reward-related signals in cerebellar inputs[60] and outputs[61–63] during motor control tasks. We suggest that these interactions between the basal ganglia and the cerebellum forms the minimal mechanism of the metacognitive process to monitor and control the error minimization process in implicit motor learning.

## Methods

### Theory: Reinforcement learning of motor learning properties

Typical models of motor adaptation[21] assume that when a learner generates motor commands $m^{(k)}$ on trial $k$, the perturbation $p$ biases the hand trajectory $y^{(k)} = m^{(k)} + p$. Meanwhile, the learner predicts the hand position $\hat{y}$ with the memory of the perturbation $x^{(k)}$ with $\hat{y}^{(k)} = m^{(k)} + x^{(k)}$. The perturbation creates a sensory prediction error $e^{(k)} = y^{(k)} - \hat{y}^{(k)} = p - x^{(k)}$. Following the conventional formulation of motor adaptation[10,11,17], when the perturbation $p$ is given to the learner, the memory $x^{(k)}$ on trial $k$ is updated by the sensory prediction errors $e^{(k)}$ with learning rate $\beta$ and is retained with retention rate $\alpha$:

$$\begin{cases} x^{(k+1)} = \alpha x^{(k)} + \beta e^{(k)} + n_x^{(k)}, \\ e^{(k)} = p - x^{(k)}, \end{cases} \quad (1)$$

where $n_x^{(k)}$ is a memory update noise with zero mean and variance $\sigma_x^2$.

This formulation is useful to describe error-based learning in multiple experimental paradigms and is well established in the motor learning literature[8,11,21]. However, in order to consider a meta-learning process, we adopt a more general framework for memory update, which leads to Eq. (1) under simple assumptions. We first assume that memory update is achieved by an action $u$ in the memory space, such that

$$x^{(k+1)} = x^{(k)} + u^{(k)}, \quad (2)$$

where the memory update (action) $u^{(k)}$ is generated from a normal distribution, spanning the space of $x^{(k)}$ and $e^{(k)}$,

$$u^{(k)} \sim \pi_{\boldsymbol{\theta}}\left(u^{(k)}|x^{(k)}, e^{(k)}\right) = \mathcal{N}\left(f_{\boldsymbol{\theta}}\left(x^{(k)}, e^{(k)}\right), \sigma_x^2\right), \quad (3)$$

which is called an action policy, and is characterized by a parameter vector $\boldsymbol{\theta}$ of a function approximator $f_{\boldsymbol{\theta}}(x^{(k)}, e^{(k)})$. In this framework, the memory update is considered a Markov Decision Problem, where the memory on the next trial $x^{(k+1)}$ is determined by the memory on the current trial $x^{(k)}$ and the action $u^{(k)}$. If we consider a policy specific to a motor learning problem implemented with the linear function $f_{\boldsymbol{\theta}} = \theta_1 x^{(k)} + \theta_2 e^{(k)}$ with $\boldsymbol{\theta} = [\theta_1, \theta_2]^T$, this formulation becomes equivalent to Eq. (1) when $\theta_1$ is replaced with $\alpha - 1$ and $\theta_2$ with $\beta$.

We next assume that the long-term goal for a learning agent with policy $u^{(k)} \sim \pi_{\boldsymbol{\theta}}(u^{(k)}|x^{(k)}, e^{(k)})$ is to maximize expected rewards (and minimize expected punishment) $r(x^{(k+1)})$ caused by the memory update. Thus, the learner's objective function for motor learning between $k$ and $k+1$ is $J(\boldsymbol{\theta}) = E_{\pi_{\boldsymbol{\theta}}}(r(x^{(k+1)}))$. We take a gradient of this objective function along the parameter $\boldsymbol{\theta}$ to obtain the direction of its update,

$$\Delta\boldsymbol{\theta} \sim \nabla_{\boldsymbol{\theta}} J(\boldsymbol{\theta}), \quad (4)$$

where $\nabla_{\boldsymbol{\theta}} J(\boldsymbol{\theta})$ is approximated by

$$\nabla_{\boldsymbol{\theta}} \log \pi_{\boldsymbol{\theta}} \cdot r(x^{(k+1)}), \quad (5)$$

in accordance with the policy gradient theorem[18].

Since the policy is Gaussian, the log of the policy function is given by:

$$\log \pi_{\boldsymbol{\theta}}(u^{(k)}|x^{(k)}, e^{(k)}) = -\frac{1}{2\sigma_x^2}(f_{\boldsymbol{\theta}}(x^{(k)}, e^{(k)}) - u^{(k)})^2 + const. \quad (6)$$

Then, the partial derivative of this log policy function is

$$\nabla_{\boldsymbol{\theta}} \log \pi_{\boldsymbol{\theta}}(u^{(k)}|x^{(k)}, e^{(k)}) = \frac{1}{\sigma_x^2}\left(u^{(k)} - f_{\boldsymbol{\theta}}\left(x^{(k)}, e^{(k)}\right)\right)\frac{\partial f_{\boldsymbol{\theta}}}{\partial \boldsymbol{\theta}}. \quad (7)$$

Rewriting Eq. (3) as $u^{(k)} = f_{\boldsymbol{\theta}} + n_x^{(k)}$, we obtain a gradient descent (or ascent) rule for the policy $u^{(k)} \sim \pi_{\boldsymbol{\theta}}(u^{(k)}|x^{(k)}, e^{(k)})$,

$$\begin{aligned} \boldsymbol{\theta}^{(k+1|k)} &= \boldsymbol{\theta}^{(k|k)} + \frac{\eta}{\sigma_x^2} \cdot \left(u^{(k)} - f_{\boldsymbol{\theta}}\left(x^{(k)}, e^{(k)}\right)\right)\frac{\partial f_{\boldsymbol{\theta}}}{\partial \boldsymbol{\theta}} \cdot r\left(x^{(k+1)}\right) \\ &= \boldsymbol{\theta}^{(k|k)} + \frac{\eta}{\sigma_x^2} \cdot n_x^{(k)} \cdot r\left(x^{(k+1)}\right) \cdot \frac{\partial f_{\boldsymbol{\theta}}}{\partial \boldsymbol{\theta}}, \end{aligned} \quad (8)$$

where $\eta$ is the learning rate of reinforcement learning, that is, the meta-learning rate. Thus, after the memory update is conducted at trial $k$ in the lower motor learning layer, the parameter used for the memory update $\boldsymbol{\theta}^{(k|k)}$ is updated to $\boldsymbol{\theta}^{(k+1|k)}$ by the meta-learning rule (Eq. (8)) in the higher meta-learning layer. This updated meta-parameter $\boldsymbol{\theta}^{(k+1|k)}$ is used for motor learning at the next step $k+1$ after shifting $\boldsymbol{\theta}^{(k+1|k)}$ to $\boldsymbol{\theta}^{(k+1|k+1)}$.

For the linear function $f_{\boldsymbol{\theta}} = \theta_1 x^{(k)} + \theta_2 e^{(k)} = (\alpha - 1)x^{(k)} + \beta e^{(k)}$, we have $\frac{\partial f_{\boldsymbol{\theta}}}{\partial \alpha} = x^{(k)}$, $\frac{\partial f_{\boldsymbol{\theta}}}{\partial \beta} = e^{(k)}$. The meta-learning rules for $\alpha$ and $\beta$ are therefore:

$$\begin{aligned} \alpha^{(k+1|k)} &= \alpha^{(k|k)} + \frac{\eta_\alpha}{\sigma_x^2} \cdot n_x^{(k)} \cdot r(x^{(k+1)}) \cdot x^{(k)} \\ \beta^{(k+1|k)} &= \beta^{(k|k)} + \frac{\eta_\beta}{\sigma_x^2} \cdot n_x^{(k)} \cdot r(x^{(k+1)}) \cdot e^{(k)}. \end{aligned} \quad (9)$$

Thus, the meta-parameters of motor learning $\alpha$ and $\beta$ are updated by the exploration noise $n_x^{(k)}$ and the reward given by the updated memory $r(x^{(k+1)})$ with influence by the memory $x^{(k)}$ and the prediction error $e^{(k)}$, respectively. Therefore, if the reward is externally manipulated such that more motor learning provides larger rewards, reinforcement learning increases $\alpha$ and $\beta$. In contrast, if it is manipulated such that less learning provides larger rewards, it decreases those parameters. Note that the reward function is unbeknown to learners,

and a goal is to maximize reward through trial-and-error without knowing the function.

In computer simulations, the same number of learners was used as in the actual experiment ($N = 20$ per group for each experiment) to calculate the mean and variance of the profiles of $\alpha$, $\beta$, and $x$ in Fig. 1E. Free parameters were manually set as follows: $[\alpha^{(1|1)}, \beta^{(1|1)}, \eta_\alpha, \eta_\beta, \sigma_x] = [0.97, 0.4, 4.9 \times 10^{-5}, 4.0 \times 10^{-4}, 0.75]$ so that they approximately matched the human participants' parameters estimated in the section below. The score function and the sequence of meta-learning training were as in the experiments (Fig. 1D–F). The range of the parameter ($\alpha$, $\beta$) was constrained between 0 and 1 by setting $\eta_\alpha = 0$ when $\alpha \le 0$ or $\alpha \ge 1$ and $\eta_\beta = 0$ when $\beta \le 0$ or $\beta \ge 1$.

We briefly note here that our theory is not limited to the linear learning policy $f_\theta = \theta_1 x^{(k)} + \theta_2 e^{(k)}$. If we consider a general nonlinear network function $f_\theta = \theta^T G(e, x)$ with basis function $\mathbf{G}(e, x)$ spanning the space of $x^{(k)}$ and $e^{(k)}$, the meta-learning effect is predicted to generalize over the space $x$ and $e$ that the learner experienced during the training, which is characterized by the shape of the basis function $\mathbf{G}(e, x)$[8].

## Participants

Forty right-handed participants without a history of neurological or motor disorders volunteered for Experiment 1, and forty participants for Experiment 2 (in total 44 males; aged 18–30 years, the mean is 21 years old). Participant right-handedness was confirmed using the Edinburgh Handedness Inventory. They were paid 2,150 JPY for their participation, with additional performance-based compensation of up to 1,000 JPY.

## Task design

**General.** Participants performed the task using a robot manipulandum that moved only in the horizontal plane[64]. They sat on a chair and held the robot handle in their right hands. A horizontal mirror covered the task space, occluding the hand and forearm. A monitor was placed above the mirror, and participants looked at visual stimuli presented on the monitor, which were reflected in the mirror[64]. The height between the handle and the mirror was the same as the height between the mirror and the monitor, such that the visual stimuli appeared at the same height as the handle. The manipulandum was self-made and controlled by Python 3.7.9 and LabView 2019.

**Trial flow.** We modified the flow of trials and manipulation of feedback of a previously established task to control both the prediction error (the error between the predicted hand movement and the cursor feedback) and the task error (the error between the visual target and the cursor)[9]. Participants first positioned their hand cursor (a white circle 5 mm in diameter) at a starting point (a gray circle 9 mm in diameter) at the bottom center of the screen. Participants made a rapid shooting movement after a target (a 5-mm blue circle) blinked for 100 ms, 10 cm from the starting point (Fig. S1). After initiation of the shooting movement, the target reappeared again when the hand reached 8 cm. To avoid use-dependent learning[65], the target direction was pseudo-randomly selected from 1 of 7 directions: $-15°$, $-10°$, $-5°$, $0°$ (right in front of the participant), $5°$, $10°$, and $15°$. The counter-clockwise direction was defined as positive in angular coordinates. To maintain similar kinematics across trials, "Too Fast" or "Too Slow" was displayed as a warning when movement duration was <150 ms or >250 ms, respectively. In addition, to discourage predictive movement initiation, the waiting time at the start point was randomized between 800 and 1200 ms. To further discourage predictive movement initiation or cognitive strategy, the trial was aborted and retried if participants failed to initiate movement between 100 ms and 450 ms after target presentation.

**Trial type and task schedule.** Participants experienced four types of trials: Null, Sensory-Error (E), Reward (R), and Error-Clamp trials.

In Null trials, participants made shooting movements toward the targets with veridical online cursor feedback. E trials were identical to Null trials except that a cursor rotation (a discrepancy between the hand and the cursor) was imposed as a perturbation to induce sensory prediction error. The rotation size was manipulated among E trials as described below. In addition, the target reappeared in the direction of the cursor at 10 cm, making participants perceive the cursor crossing at the center of the reappeared visual target, which clamped the task error to zero (TE clamp). Thus, in E trials, the participant experienced a sensory prediction error, but not a task error[9]. In R trials, a numerical score as a function of the updated memory computed from the aftereffect, the angle distance from the presented target and the reach direction, was presented as a reward. Online cursor feedback was absent to remove feedback related to sensory prediction error, and the target did not reappear at the crossing. Finally, in Error-Clamp trials, the cursor direction was constrained to the target direction so that the cursor moved straight to the target regardless of hand movement, clamping both sensory error and task error to zero[27,28]. An illustration of the flow and manipulation of each trial type is shown in Fig. S1.

The task comprised two phases: meta-learning training and Probe (Fig. 1E, F). The meta-learning training phase consisted of 6 cycles of 10 Null trials (Washout) followed by five pairs of E and R trials (20 trials per cycle, 120 in total). $+5°$ rotation was used in E. Since the learner received the reward feedback in R for multiple trials in the training phase, the updated memory after the first R trial is potentially influenced by the accumulation of a reward effect. Thus, we devised the Probe phase to examine whether prior meta-learning training affects motor learning ability in a typical visuomotor rotation task where the visual rotation E is imposed consecutively in a step-perturbation manner. To measure the speed and retention of learning, the Probe phase consisted of 5 E trials with no rotation followed by 10 E trials with $+7°$ rotation and by 15 Error-Clamp trials.

The experiments comprised four blocks. The first block contained a Probe phase followed by a meta-learning training phase and another Probe phase. The remaining blocks consisted of a meta-learning training phase followed by a Probe phase. 30 Null trials and one-minute breaks were inserted before and after each Probe phase. The score was removed in R trials in the first block to measure baseline behavior without reward feedback.

**Instruction.** We modified the instructions from a previous visuomotor rotation study[9] as follows. "There may be some manipulation of the cursor and/or the target. Even if you notice any manipulation, your task remains the same: to move your hand to the target as best you can." Participants were not informed what the exact manipulations, neither were they informed of the relationship between their hand directions and the scores. They answered a post-task written questionnaire to verify that they did not aim in any direction other than toward the target. In addition, participants were informed about the presence/absence of cursor feedback and score in the Null, E, and R trials and practiced each type of trial before the task. They were also explicitly told that one point was worth 1 JPY and that the task goal was to maximize their compensation by reaching the target as accurately as possible (see details below). In Experiment 1, participants were told that the additional compensation was initially maximum (1000 JPY), but that negative scores would decrease this initial amount throughout the experiment. In Experiment 2, they were told that the additional compensation was initially zero (0 JPY) and that positive scores would increase this initial amount to up to 1000 JPY.

**Experiment groups and score calculations.** Participants were pseudo-randomly assigned to two groups in each experiment ($N = 20$ per group), **Promote** and **Suppress** (Fig. 1). The groups differed in how the score was calculated based on the motor memory, as measured by the aftereffect ($x = T - h$, Fig. 1D) and then computed the score by

$score = S \cdot x + s_0 + V$ (rounding to integers), where $S$ is the slope that determines the learning-outcome structure of the score feedback, $s_0$ determines the intercept of the score function, and $V$ is the bias of the score that determines the valence level of the score feedback. In **Promote**, $S = 1, s_0 = 0$, the score increased from 0 to 5 as the learning progressed to encourage learning. In **Suppress,** $S = -1, s_0 = 5$, and the score therefore decreased from 5 to 0 as the learning progressed to discourage learning. In Experiment 1, in order to deliver punishments, $V = -5$. The score was biased to the negative (Fig. 1D). In Experiment 2, in order to deliver rewards, $V = 5$. The score was biased to the positive. We set the maximum and minimum score to 0 in Experiments 1 and 2, respectively, so that the sign did not change during an experiment. One point corresponded to 1 JPY. Additional monetary compensation was given to participants at the end of the experiment.

## Analysis

### Estimating memory update and retention in training trials and Probe.
In meta-learning training, we used the first and last R trials as indications of initial memory update (learning) and accumulated memory update, respectively. In the Probe phases, we used both the second E trial (as the initial update occurred after experiencing sensory prediction error in the first E trial) as the indication of initial memory update and the average of all the Error-Clamp trials to examine the accumulated updates and their retention.

Specifically, after participants experienced 10 Null trials in each training cycle and 30 Null trials before Probe, the initial update, the first response to the given perturbation, reflected the memory update in response to the prediction error because the memory was approximately zero, $x^{(k+1)} = \alpha 0 + \beta e^{(k)} = \beta e^{(k)}$; thus, this initial update captured the change in the learning rate $\beta$. Conversely, the last trials of training captured the accumulated effect of change in both $\alpha$ and $\beta$. In the Error-Clamp trials of Probe, because the error was clamped at zero, $e^{(k)} = 0$, $x^{(k+1)} = \alpha x^{(k)} + \beta 0 = \alpha x^{(k)}$, we measured the change in $\alpha$.

To minimize the effect of feedback correction during the movement, the hand direction was measured when the reaching distance exceeded 5 cm from the starting point, which was on average about 100 ms after movement initiation. However, because the reaching movement was not perfectly straight, the hand direction 5 cm from the start point resulted in a small bias compared to the hand direction at the endpoint. The average bias measured in the baseline trials of Block 1 was subtracted from all reach direction data to capture the change of memory.

### Statistical analysis of measured behavior.
To evaluate meta-learning effects, we performed linear mixed-effect model analyses[66] for each experiment to estimate the slope of change over blocks for the measured initial memory update, the accumulated memory update, retention, and the score performance. All models included "participant" as the random intercept effect to represent inter-participant variability. Inclusion of the random effect was validated by confirming that it led to smaller Akaike Information Criteria (AIC)[67]. The random slope effect was not included, so as to avoid redundancy and failure of the estimation to converge. For simplicity, random effects were omitted in the equations below.

To test the effects of the Learning-Outcome Structure on training, we estimated the interaction between the binary variable LOS (**Promote** = 1 and **Suppress** = 0) and the continuous Block number. Thus, for the initial memory update, the accumulated memory update, and score, the equation was

$$d \sim \gamma_0 + \gamma_1 LOS + \gamma_2 Block + \gamma_3 LOS \cdot Block, \quad (10)$$

where $d$ was either the initial memory update, the accumulated memory update, or the score. $\gamma_1$ represents an immediate constant effect of LOS; $\gamma_2$ represents how the training gradually updates

learning performance in the "reference" group, i.e., **Suppress** since $LOS = 0$; $\gamma_3$ represents the interaction between LOS and Block which indicates how LOS's influence on $d$ develops over Block, i.e., the meta-learning effect. We confirmed no significant differences between **Promote** and **Suppress** on the reach direction ($T-h$) in the baseline trial before participants responded to the intial pertubation (11th cycle trial in meta-learning training and 6th trial in Probe) in Blocks 1 and 4 for each experiment, as shown in Figs. 2A, D, 3A, D.

### Estimation and statistical analysis of learning parameters from Probe data.
The purpose here was to estimate the evolution of the retention rate $\alpha$ and learning rate $\beta$ of the motor learning model. Motor learning was described by a stochastic state-space model (Eq. (1)) and the memory change was measured via the change of the reach direction corrupted by observation noises,

$$x^{(k+1)} = \alpha x^{(k)} + \beta e^{(k)} + n_x^{(k)}, \text{(state update)}$$
$$e^{(k)} = p - x^{(k)}, \text{(sensory prediction error)} \quad (11)$$
$$(T - h) = y = x^{(k)} + n_y^{(k)}, \text{(experimenter's observation)}$$

where $n_x^{(k)} \sim \mathcal{N}(0, \sigma_x^2)$ is the noise in the memory update process with zero on mean and $\sigma_x^2$ on the variance and where $n_y^{(k)} \sim \mathcal{N}(0, \sigma_y^2)$ is the observation noise due to the motor noise and the experimental setup noise with zero on mean and $\sigma_y^2$ on the variance. We estimated $\alpha, \beta$ and their meta-learning effects while estimating the magnitude of the state update noise and the observation noise ($\sigma_x, \sigma_y$). We further assumed that reach direction was influenced by both the within-participant variance and across-participant variance. Here, we incorporated the participant random intercept into the model (Eq. (11)) and avoided a two-step approach where the participant-level estimates of the parameters were obtained first, and subsequently the follow-up group-level statistical tests were conducted[68]. This contributes to having accurate group-level estimates of the parameters of interest, similar to the mixed-effect model analysis applied in the analyses above. Thus, the integrated model was

$$x_{[s,block]}^{(k+1)} \sim \mathcal{N}\left(\alpha_{[g,block]}x_{[s,block]}^{(k)} + \beta_{[g,block]}e_{[s,block]}^{(k)}, \sigma_x^2\right),$$
$$e_{[s,block]}^{(k)} = p - x_{[s,block]}^{(k)}, \quad (12)$$
$$y_{[s,block]}^{(k)} \sim \mathcal{N}\left(x_{[s,block]}^{(k)} + r_s, \sigma_y^2\right),$$

where the subscript $s$ denotes participant, the subscript $g$ denotes group, $r_s$ is a random factor of observation (individual bias of the reach), and learning parameters $\alpha_{[g,block]}, \beta_{[g,block]}$ are constant throughout each block, but differ between groups. The meta-learning effect, i.e., the block effect of each group (**Promote/Suppress**) was modeled by the following linear function of Block,

$$\alpha_{[g,block]} = \alpha_{[g,base]} + \gamma_g^\alpha \cdot Block$$
$$\beta_{[g,block]} = \beta_{[g,base]} + \gamma_g^\beta \cdot Block, \quad (13)$$

where $\alpha_{[g,base]}, \beta_{[g,base]}$ are the baseline block's retention and learning rates and the slope of the Block effect.

To estimate group level distributions of $\gamma_g^\alpha, \gamma_g^\beta, \alpha_{[g,base]}$, and $\beta_{[g,base]}$, while estimating $r_s$ for each participant and $\sigma_x, \sigma_y$ common across all participants, we performed parameter estimation by Bayesian Inference. We used Markov chain Monte Carlo (MCMC)[29,30], as implemented in the cmdstanr package[69], which provides the exact posterior distribution of these parameters of interests ($\gamma_g^\alpha, \gamma_g^\beta, \alpha_{[g,base]}, \beta_{[g,base]}$) of each group.

We determined the prior distribution following recommended procedures[70]. We avoided non-informative flat priors and adopted weakly informative priors, which can stabilize the estimation. In

addition, we incorporated a minimum amount of domain-specific knowledge of the task design. Because we did not have any information about the slopes $\gamma_g^\alpha, \gamma_g^\beta$, we set weakly informative priors centered at zero: $\gamma_g^\alpha \sim \mathcal{N}(0,(0.5)^2)$, $\gamma_g^\beta \sim \mathcal{N}(0,(0.5)^2)$. According to the task design, the memory takes between 0° and 7°, and the initial memory should be zero. Thus, we set weakly informative prior for the initial value of memory, $x_{[s,block]}^{(1)} \sim \mathcal{N}(0,(5)^2)$, random factors for participants, $r_s \sim \mathcal{N}(0,(5)^2)$, and the size of noise $\sigma_x \sim \exp(1)$, $\sigma_y \sim \exp(1)$. The learning parameters $\alpha$ and $\beta$ are between 0 and 1, according to previous motor learning studies[8,11,21,23]. Thus, we set the baseline learning parameters as $\alpha_{[g,base]} \sim \mathcal{N}(0.5,1)$, $\beta_{[g,base]} \sim \mathcal{N}(0.5,1)$. We also confirmed that comparable estimation results were obtained when non-informative flat priors were used for $\alpha_{[g,base]}$ and $\beta_{[g,base]}$, which validated the contribution of the likelihood of the measured data points.

In addition, to determine the profiles of the evolution of $\alpha_{[g,block]}$ and $\beta_{[g,block]}$ (Figs. 2F and 3F, left panels), these parameters were estimated for each block and group in a separate model without assuming linear changes. That is, instead of estimating the slopes ($\gamma_g^\alpha, \gamma_g^\beta$), each $\alpha_{[g,block]}$ and $\beta_{[g,block]}$ were directly estimated like $\alpha_{[g,base]}$ and $\beta_{[g,base]}$. The priors for $\alpha_{[g,block]}$ and $\beta_{[g,block]}$ were $\alpha_{[g,block]} \sim \mathcal{N}(0.5,1)$, $\beta_{[g,block]} \sim \mathcal{N}(0.5,1)$. Comparable results were obtained again when non-informative flat priors were used for $\alpha_{[g,block]}$ and $\beta_{[g,block]}$

Posterior density distributions were derived from four chains of 8000 sampling per estimation. The initial half of the sampling (4000 samples) was discarded as warm-up, and the last half of the samples (4000 samples) was used to compute high-density intervals (HDI). Convergence criteria for MCMC were set at the Gelman-Rubin convergence statistics $R\text{-}hat < 1.05$[29]. A parameter was considered significant when the 95% HDI did not included 0. In addition, we calculated the proportion of samples above/below zero to the total number of samples (1 - proportion of direction), corresponding to p-values. We confirmed that this method could estimate reasonable means and 95% HDI for the estimated $\sigma_x$ (mean = 0.75, HDI = [0.70, 0.80]) and $\sigma_y$ (mean = 2.15, HDI = [2.12, 2.19]). We set the variances in simulations (Fig. 1E) based on these mean values.

**Estimation and statistical analysis of meta-learning rates.** Estimating the meta-learning rates ($\eta_\alpha, \eta_\beta$) provides direct evidence of reinforcement learning for meta-learning. However, estimating these rates for each participant is challenging because, according to Eq. (8), meta-learning behavior is determined by each participant's trial-to-trial series of memory noise $n_x^{(k)}$, which is latent. A difficulty is that estimation of $n_x^{(k)}$ from the data is not possible. Indeed, whereas $\hat{x}^{(k)}$ can be estimated via Bayesian estimation using Eq. (12), and subtracting $\hat{x}^{(k)}$ from the measured hand direction $(T-h)$ provides the residual $n_x^{(k)} + n_y^{(k)}$, estimating $n_x^{(k)}$ from this residual is not possible because $n_y^{(k)}$ is also unknown.

Therefore, we estimated these parameters from the averaged meta-learning effect in the training trials. Equation (9) determines the size of the update of retention rate:

$$\Delta\alpha^{(k|k)} = \frac{\eta_\alpha}{\sigma_x^2} \cdot n_x^{(k)} \cdot r(x^{(k+1)}) \cdot x^{(k)}. \tag{14}$$

The expectation of this update is

$$E(\Delta\alpha^{(k|k)}) = E\left(\frac{\eta_\alpha}{\sigma_x^2} \cdot n_x^{(k)} \cdot r(x^{(k+1)}) \cdot x^{(k)}\right). \tag{15}$$

As a reminder, the score is the linear function of memory $r(x^{(k+1)}) = S \cdot x^{(k+1)}$ in the range, $0 < x^{(k+1)} < 5$, where $S = 1$ for **Promote** condition and $S = -1$ for **Suppress** condition, except Block 1 (baseline

block) where $S = 0$. Thus, we have

$$E(\Delta\alpha^{(k|k)}) = \frac{\eta_\alpha}{\sigma_x^2} \cdot S \cdot E(n_x^{(k)} \cdot x^{(k+1)} \cdot x^{(k)}). \tag{16}$$

By substituting $x^{(k+1)}$ with $\alpha x^{(k)} + \beta e^{(k)} + n_x^{(k)}$, we obtain

$$\begin{aligned} E(\Delta\alpha^{(k|k)}) &= \frac{\eta_\alpha}{\sigma_x^2} \cdot S \cdot E(n_x^{(k)} \cdot (\alpha x^{(k)} + \beta e^{(k)} + n_x^{(k)}) \cdot x^{(k)}) \\ &= \eta_\alpha \cdot S \cdot E(x^{(k)}). \end{aligned} \tag{17}$$

Notably, the memory noise $n_x^{(k)}$ is averaged out and variance $\sigma_x^2$ is canceled; thus, the issue of the dependency of the amount of meta-learning on the memory noise was solved. Accordingly, $E(x^{(k)})$ can be approximated from the observed reach directions during the training trials without estimating the sequence of $n_x^{(k)}$. To obtain robust and population-level estimates, we approximated $\bar{x}^{(c,p)} = E(x^{(c,p)})$ by the across-participants average of reach behavior (Figs. 2A, 3A), $\bar{x}^{(c,p)} = \frac{1}{N} \sum_{s=1}^{N} (T-h)^{(s,c,p)}$, where the superscript $p$ is the trial pair index, $c$ is the cycle index, $s$ is the participant index, $N$ ($= 20$) is the total number of participants in each group. Since each participant experienced six cycles of meta-learning training with the same perturbation pattern composed of 5 repetitions of SR trial pairs in each training block, the expectation of the meta-learning effect on memory retention of each block is

$$\Delta\bar{\alpha}_{[g,block]} = \eta_\alpha \cdot S \cdot \sum_{c=1}^{6} \sum_{p=1}^{5} \bar{x}_{[g,block]}^{(c,p)}. \tag{18}$$

Similarly, the expectation of the meta-learning effect on the learning rate of each block is

$$\Delta\bar{\beta}_{[g,block]} = \eta_\beta \cdot S \cdot \sum_{c=1}^{6} \sum_{p=1}^{5} \bar{e}_{[g,block]}^{(c,p)}. \tag{19}$$

This suggests that updates of $\alpha$, $\beta$, on average, are determined by the meta-learning rates $\eta_\alpha, \eta_\beta$, the score slope $S$, and the average of the generated reach $\bar{x}_{[g,block]}^{(c,p)}$ and error $\bar{e}_{[g,block]}^{(c,p)}$. Because $\bar{x}_{[g,block]}^{(c,p)}$ and $\bar{e}_{[g,block]}^{(c,p)}$ can be computed from the measured hand direction data $(T-h)$, we incorporated $\Delta\bar{\alpha}_{[g,block]}$ and $\Delta\bar{\beta}_{[g,block]}$ into the MCMC estimation algorithm with Eq. (12) by replacing Eq. (13) with

$$\begin{aligned} \alpha_{[g,block]} &= \alpha_{[g,base]} + \sum_{b=1}^{block} \Delta\bar{\alpha}_{[g,b]}, \\ \beta_{[g,block]} &= \beta_{[g,base]} + \sum_{b=1}^{block} \Delta\bar{\beta}_{[g,b]}. \end{aligned} \tag{20}$$

Thus,

$$\begin{aligned} \alpha_{[g,block]} &= \alpha_{[g,base]} + \eta_\alpha \cdot S \cdot \sum_{b=1}^{block} \sum_{c=1}^{6} \sum_{p=1}^{5} \bar{x}_{[g,b]}^{(c,p)}, \\ \beta_{[g,block]} &= \beta_{[g,base]} + \eta_\beta \cdot S \cdot \sum_{b=1}^{block} \sum_{c=1}^{6} \sum_{p=1}^{5} \bar{e}_{[g,b]}^{(c,p)}. \end{aligned} \tag{21}$$

Then, we estimated the posterior distribution of $\eta_\alpha$, $\eta_\beta$ for each experiment (Experiment 1 for negative valence and Experiment 2 for positive valence). Note that, since $\eta_\alpha$ and $\eta_\beta$ were expected to be much smaller than other parameters such as $\alpha$ and $\beta$, according to our simulation studies (Fig. 1E), normalization of these parameters was critical to stabilizing the MCMC search. We therefore inserted a scaling parameter $k_e = 100$ and estimated the group distribution of $k_e\eta_\alpha, k_e\eta_\beta$ by MCMC and then reported the estimated $\eta_\alpha, \eta_\beta$ by dividing them by

$k_e = 100$. The weakly informative priors for these values were $k_e \eta_\alpha \sim \mathcal{N}(1,1)$, $k_e \eta_{\beta,} \sim \mathcal{N}(1,1)$. Other priors were as in the previous section. We also confirmed that comparable estimation results were obtained with less informative priors, $k_e \eta_\alpha \sim \mathcal{N}(0,3)$, $k_e \eta_{\beta,} \sim \mathcal{N}(0,3)$. Again, convergence was checked by R-hat < 1.05.

**Statistical package, coding, and other notes.** All data processing and statistical analyses were performed in R version 4.0.2 using the lm, lme4[66], lmerTest[71], margins[72], and cmdstanr[69] packages. For the linear mixed-effect model analyses, dummy coding was used to represent the categorical variables (Learning-Outcome Structure: **Promote** as 1 and **Suppress** as 0. Valence: Positive as 1 and Negative as 0), and the numerical variable (Block) was treated as a continuous variable, with the initial value aligned to 0. All tests were performed as two-sided tests, and the boundary for the p-value was set to 0.05 to determine statistical significance.

## Simulation of previous studies

We performed a series of simulations to demonstrate that the proposed Meta-learning model can replicate previous reports on changes in motor learning speed by the history of sensory prediction error[8], the reward/punishment on trajectory error[11], and manipulation of the task error[9]. To do so, we assumed that the task error functions as punishing performance feedback, as discussed in the main manuscript. This assumption was based on the evidence that the task error induces activity in the striatum[23], a crucial neural region for reinforcement learning, which shows reward/value-related activities[39].

These three papers used motor learning models similar to Eq. (1). In the force field adaptation task of Herzfeld et al.[8], by using the simulated motor memories with Eq. (1), we predicted the experimentally measured error sensitivity $\delta_{ex}$ by the probe trial designed in Herzfeld et al.[8], which uses a pair of measured forces before and after the force perturbation: $\delta_{ex} = \frac{\mu^{(k+1)} - \alpha^2 \mu^{(k-1)}}{e^{(k)}}$ where $\mu^{(k)}$ [a.u.] is the motor memory which corresponds to $x^{(k)}$ in Eq. (1) and $e^{(k)}$ is the sensory error [cm] in the force perturbation trial. For our simulation, since we can assume that, after enough washout trials, the memory formed in the training block had decayed to zero before the probe trial, without loss of generality, we set $\mu^{(k-1)} = 0$ and then $\delta_{ex} = \mu^{(k+1)}/e^{(k)}$. In the force field adaptation task, the sensory error is scaled by the biomechanical property $e^{(k)} = D \cdot v_y \cdot B \cdot (p^{(k)} - \mu^{(k)}) = C \cdot (p^{(k)} - \mu^{(k)})$ [cm], where $D$ [cm/N] is the compliance, $v_y$ is the movement speed [m/s], and $B$ [N s/m] is viscosity. In Herzfeld et al.[8], $B$ was set 13 [N s/m]. According to the previously estimated value $D$ takes approximately 0.3 [cm/N][54]. The movement speed is typically 0.4–0.5 m/s. Thus, we set $C = D \cdot v_y \cdot B = 2.0$ as an approximated value, which roughly matches the size of the deviation (~2 cm) from the straight path in the reported hand trajectories in Herzfeld et al.[8].

Because $e^{(k)}$ represented an error in the physical space, it was proportional to the task error, except for the RandomTE and NoTE conditions in Leow et al.[9], where the task error was random and clamped to 0, respectively. Given our assumption that the task error serves as punishing feedback, we defined reward feedback $r$ for Herzfeld et al.[8] and Leow et al.[9] as

$$r^{(k)} = -|TaskError^{(k)}| \tag{22}$$

In Galea et al.[11], a reward/punishment signal was given in addition to the task error; therefore, we assumed for simplicity that reward feedback is a sum of task error and score with scaling constants, as follows:

$$r^{(k)} = -c_{TE}|TaskError^{(k)}| + c_s score^{(k)}. \tag{23}$$

Again, reinforcement learning agents maximize reward through trial-and-error without knowing the relationship between memory and reward (i.e., the reward function).

We hypothesized that the effect of reward/punishment on trajectory error found in Galea et al.[11] is mediated by the meta-learning rates, which can be modeled by different $\eta_\alpha$, $\eta_\beta$ for reward and punishment conditions (valence effect). Specifically, in Galea et al.[11], smaller $\eta_\alpha$ and larger $\eta_\beta$ were used in the punishment condition compared to the reward conditions (i.e., $\eta_{\alpha-pun} < \eta_{\alpha-rwd}$, $\eta_{\beta-pun} > \eta_{\beta-rwd}$) to achieve faster learning with punishment and stronger retention with reward. Free parameters in each condition and study are manually set and summarized in Table S16. Note that comparable results are obtained when the free parameters are tuned through an optimization algorithm (See Fig. S5D and Table S19 in Supplementary Note 2)

Task schedules in the simulations were matched to those in previous studies, except Galea et al.[11], where one epoch was treated as one trial. To simulate forgetting, we assumed it was equivalent to 6 epochs of reaching under No vision (which was omitted in Fig. 4F). Note that this assumption did not influence learning parameters or subsequent behavior, because neither task error nor score was presented during No vision (i.e., $r = 0$). The numbers of simulated agents were 10 per condition for Herzfeld et al.[8], 200 for Galea et al.[11], and 100 for Leow et al.[9].

## Reporting summary

Further information on research design is available in the Nature Portfolio Reporting Summary linked to this article.

## Data availability

Source data are provided with this paper and are available as an Excel document through Figshare (https://doi.org/10.6084/m9.figshare.22067054).

## Code availability

Stan model text files, R scripts for the simulations, and data files are available through Figshare (https://doi.org/10.6084/m9.figshare.22067054). Scripts for the manipulandum are not included because they are solely for operation of our self-made, non-commercialized manipulandum. They may be still available upon reasonable requests to the corresponding author.

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

## Acknowledgements

This work was supported by the Japan Society for the Promotion of Science KAKENHI (JP19H04977, JP19H05729, and JP22H00498). T.S. was supported by a JSPS Research Fellowship for Young Scientists and KAKENHI (JP19J20366). N.S. was supported by NIH R21 NS120274.

## Author contributions

Conceptualization: T.S., N.S., J.I.; Methodology: T.S., J.I.; Theoretical model design: T.S., J.I.; Data curation: T.S.; Investigation: T.S., J.I.; Visualization: T.S.; Funding acquisition: J.I.; Writing—original draft: T.S., J.I.; Writing—review & editing: T.S., N.S., J.I.

## Competing interests

The authors declare no competing interests.
