## [Peer Review File · Nature Communications]

Reinforcement learning establishes a minimal metacognitive process to monitor and control motor learning performance.REVIEWER COMMENTS

Reviewer #1 (Remarks to the Author):

The study by Sugiyama, Schweighofer, and Izawa aims to establish that reinforcement learning works as a metacognitive process to affect motor learning. They formulated a hierarchical model that uses reinforcement to modulate learning and retention rate and confirmed with empirical evidence that monetary reward can affect people's motor adaptation during training and interleaved learning probe sessions. Furthermore, positive and negative rewards affect motor learning and retention differently, consistent with previous work. To support the new model as a unified theory for motor learning, the authors also provide model simulations that qualitatively resemble the major findings of three previous motor learning research studies. [L]
[SEP]

Overall, the paper is well written with a high-quality data presentation. The two experiments are beautifully designed to show that reinforcement learning indeed affects motor learning after the training period, critical evidence for a meta-learning effect. However, this effect is weak given that the probe session is interleaved with the training session, and the effect size is small (see below). The model is also reasonable.

However, I have two major concerns regarding the new contribution of the present work.

1) The theoretical novelty about the effect of reward on up and down-regulate adaptation.

The present model assumes that reinforcement hierarchically affects the low-level adaptation by regulating its learning rate and retention. The authors pointed out that they provide "the first empirical evidence for the existence of a reinforcement learning mechanism for motor learning policy in the human brain." But the fact that reinforcement changes motor learning policy (e.g., reflected as learning rate or retention) is a well-taken idea in the field (e.g., work done by Galea and colleagues in recent years). Or, we can step back and say that the present study provides evidence that reinforcement learning works hierarchically as a meta-learning process for low-level learning processes (i.e., the processes based on sensory prediction error, SPE). But the hierarchical learning structure has also been proposed before. Kim and Ivry (2019 eLife paper, which the current paper should quote) showed that target error (TE) serves as a reinforcement signal to modulate the learning rate in an SPE-based learning task. They even proposed an alternative non-hierarchical structure for the effect of reinforcement on adaptation, i.e., TE-based reinforcement learning and SPE-based learning work in parallel to drive learning. Their data sets could not differentiate these two model alternatives.

I really like the author's emphasis on the "metacognitive process" in motor learning. In fact, besides reward signals shown here, other metacognitive processes also affect motor learning, including contextual inference, conceptual learning, or simply task instructions. These results reflect that motor learning is not just implicit, non-verbal, and procedural learning but also incorporates explicit, conscious,

and declarative components. But the paper frames its contribution or novelty by over-simplification or neglect of previous work. For example, it states that the hypothetical goal of motor learning is error minimization (Line245). This is too much of a simplification of the field: starting from Krakauer et al. 2006 seminar paper on unintentional adaptive drift based on SPE, the area already takes a multiplicative view of motor adaptation: it contains multiple components, including minimization of different errors even sacrificing task performance, reinforcement learning, use-dependent learning, etc. Similarly, when the authors claim that previous theories for learning speed regulations only consider properties of errors (Line236), the studies on reward or use-dependent learning are neglected. Thus, it is questionable whether the current study provides a conceptual step-up in the realm of motor learning as compared to similar studies, say, work done by Galea and Ivry groups in recent years.

2) Whether the presented model can provide a unified theory to account for various previous results should also be questioned.

The authors showed model simulations that qualitatively resemble the findings in three previous studies. For the first two studies (Herzfeld-2014 and Leow-2020), the current study assumes that task error serves as negative reinforcement feedback. This treatment should be cautioned, though. Target error has been proposed to drive explicit strategic learning (McDougle-2015) and implicit learning (Albert-2022). Indeed, the current paper tried to avoid strategic learning by using small perturbations (5 and 7 degrees rotation) and to focus on SPE-based implicit learning. However, having TE as a reinforcement signal is a conceptual change for the field. Thus, the authors should provide adequate evidence to support this new conceptualization of TE. The evidence here is still qualitative, based on model simulations for three previous data sets. First, there is no testing of alternative models, e.g., the Albert-2022 model; there is no direct evidence based on the new experiment (see below). Second, calling TE as negative reinforcement does not differ from the explanation offered by the original studies. For example, the first modeled study here (Herzfeld-2014) has its own modeling, which in spirit is the same as the meta-learning model: the consistency of perturbation affects adaptation performance (TE); thus, the learning rate is changed to minimize error. The reinforcement model here asserts almost the same but using different wording: the task error serves as punishment which modulates the learning rate when the consistency of perturbation changes. A similar reconceptualization of TE is used in simulating the results of Leow-2020. To convincingly “re-formulate” TE as punishment instead of as a straightforward performance error to be minimized, the authors should provide direct evidence.

Note the two new experiments presented here did not model TE (in fact, it is clamped to zero) as reinforcement but rather used monetary reward to emulate the reinforcement effect. This design cleverly dissociates the sensory prediction error and the reward in adjacent trials to show the meta-learning effect; the natural consequence is that it cannot serve as evidence for the role of TE as a reinforcement. Given that the role of TE is hotly debated with multiple models available, we shall be cautioned when the new theorization is only supported by qualitative analysis. If the meta-learning model is indeed a persimmons model, we would expect to see at least some kind of model comparison to show its superiority over other competing models.

Minor: ^[L]_[SEP]

1) The effect of meta-learning is mainly reflected in initial update and retention during the Probe task (Figure 2E). The initial update is only a point estimate of the learning rate since it is merely the response amplitude to the first perturbation. Another concern is that the retention is an aggregation of the memory decay process as it is the average of hand direction during the whole error-clamp session. This retention (Figure 2E) is not comparable to the retention rate (Figure 2F) since the former reflects both the learning extent (before decay) and the decay rate while the latter is the rate itself. ^[L]_[SEP]

2) The use of 7-degree perturbation as a test of generalization is a nice design, but it is not sufficient to rule out the possibility that the reinforcement learning system recognizes the rotation (5 degrees) as the one experienced before. This is because 5 and 7 degrees are similar in amplitude. This is reflected in Figure S3 where the error sensitivity change covers a wide range of error sizes, with the 7-degree rotation in the envelope. The more convincing generalization test for a meta-cognitive model should use a rotation in the opposite direction. ^[L]_[SEP]

3) The primary task goal for the participant is to maximize reward (instantiated by the R trials). According to the authors, the interleaved S trial induced implicit learning since participants did not report a re-aiming strategy in the post-task written questionnaire. However, the authors also stated that "...participants were informed about stimuli and feedback in the Null, S, and R trials and practiced each type of trial before the task.". Thus, the participants knew rotated feedback would be given in S trials, i.e., they knew conceptually that their actions would be affected by rotations or movement direction changes. Thus, I wonder whether a reward-modulated recall of motor solutions partially causes the meta-learning effect. This possibility is corroborated by the abnormally fast learning in the Probe phase: the adaptation to 7-degree rotation is almost finished in the first trial (shoot up to 5 degrees), which is drastically different from typical error-clamp learning. Note the additional target jump would only reduce the learning instead of accelerating it.

Line 577: 10 and 30 null trials before training cycle and Probe? The ten null trials are not specified in the task design.

The group and block effects were estimated by Bayesian inference with some modeling assumptions.

What would the result be like if we just used conventional statistical methods like ANOVA? ^[L]_[SEP]

Why not use the first Probe before training as a baseline? Or even use it to estimate $\alpha_{[g, base]}$ and $\beta_{[g, base]}$?

Line652: what kind of model is used if linearity is not assumed?

Line669: trial-trial series? ^[L]_[SEP]

Line 687: trails

Line735: it is unclear what this equation means and how the errors are converted from the Herzfeld-2014 study. [L] [SEP]

Figure 1F: The first block is marked with a different color compared to the later three blocks; this is confusing. Is the first block identical to other blocks except that it has a Probe phase before?

Reviewer #2 (Remarks to the Author):

Sugiyama / Izawa review

Sugiyama and colleagues propose a theory of meta-learning in the context of motor adaptation. Specifically, they propose a principled computational model of how the learning rate and retention of motor adaptation might be altered through experience. Their model nicely accounts for previous experimental findings relating to modulation of learning, and is confirmed in two new experiments that were specifically designed to test the predictions of the model by increasing reward/punishment feedback on trials in which participants had altered their behavior by a greater amount from the preceding trial.

This is an innovative perspective on motor learning. The paper is very clearly written. The experiments are cleverly designed and well conducted. The experimental results are clear. The theory also convincingly explains a number of well-known results relating to changes of learning rate. The discussion is thoughtful and insightful. One weakness of the paper is the quite small group sizes (8 participants), but the key contrast between groups is replicated in Experiment 2. Overall, this is an impressive and insightful paper.

I have a few (relatively minor) concerns and suggestions:

A weakness is that the figures show only group-level data. There are no figures that indicate individual participant performance, so it is difficult to assess whether the results were driven by a minority of participants that had strong effects, or whether all participants behaved similarly. I think the findings are still interesting either way, but it would be informative to be able to see some individual-level data, even if this is in the supplemental materials.

It's well-established that motor learning comprises both implicit and explicit components, but the narrative and discussion are not totally clear about which component the theory and experiment are seeking to describe.

Some previous work has suggested that certain types of meta-learning only occur when participants become explicitly aware of the experimental manipulation (e.g. Manley et al., 2014). I do wonder whether participants became explicitly aware or not in this case. It doesn't seem that participants were asked about awareness of the manipulation, and in any case, such insight should not have influenced the probes.

The introduction raises the apparent paradox that humans can cognitively modulate the learning rate of an implicit process. I don't really see this as a paradox unless the authors are proposing that the meta-learning is explicit but the adaptation itself is implicit. If this is implicit modulation of implicit learning, or explicit modulation of explicit learning, then I don't see any puzzle.

line 64-71 - I would suggest being more explicit about alpha and beta here - i.e. include an update equation. At the moment, if the reader has a reasonable understanding of conventional motor learning models their meaning will be obvious, but if not then it will require consulting the methods. It's also a little odd to define update equations for alpha and beta when they haven't yet been concretely defined.

"Learn" and "No-learn" group names are slightly inaccurate and potentially confusing. Both groups learn, after all. I worry some readers might become confused by this. So the authors might consider alternative names for these groups. e.g. "Promote learning", "Suppress learning".

The reason "S" and "R" trials are called as they are "Sensory Error" and "Reward" is only made clear in the methods. It would be helpful to briefly mention this in the results also to make it easier to remember what they are. Also, I wonder if calling "S" trials "E" trials instead might be more intuitive, since the 'Sensory' part is a bit superfluous - but this is entirely a suggestion.

"reinforcement learning of motor learning" is a slightly awkward phrasing. Perhaps something like "reinforcement learning of motor learning properties/parameters", at least at some points in the text.

Point-to-Point Response to the Reviewers

Manuscript ID: NCOMMS-22-36133-T

“Reinforcement learning establishes a minimal metacognitive process to monitor and control motor learning performance.”

by Taisei Sugiyama, Nicolas Schweighofer, Jun Izawa

We sincerely appreciate the editor and reviewers' valuable and insightful comments. We have revised and improved our manuscript as suggested. In the following, we provide a point-to-point response to each question raised. Please refer the question number (e.g., R1-A1) and the line numbers of the revised manuscript (Main_clean.pdf, colour_highlighting.pdf).

Both reviewers asked how our study differs from the established theory according to which motor learning comprises both implicit and explicit components. To answer this question in the revision, we have simulated six computational models proposed in previous studies (McDougle-2015, Kim-2019, Albert-2022, Izawa-2011, Herzfeld-2014). The three models in McDougle-2015, Kim-2019, and Albert-2022 notably include motor learning driven by task error (TE) and learning rate change via rewards and punishment. Our simulation results demonstrated that our meta-learning model better explains learning profiles affected by volatility (Herzfeld-2014), valence (Galea-2015), and manipulation of task error (Leow-2020) than these previous models with TE. In addition, our model outperforms the previous reinforcement models (Kim-2019, Izawa-2011) and the history of error model (Herzfeld-2014). A summary of these simulation results is given in the revised version of the supplemental material.

In the original version, we noticed that Fig.3G was referred after referring to Fig. 4 in the sub-section “Unification of previous results in motor learning research”. To correct this error in the revised version, we added the new subsection “Estimation of meta-learning rates in both Experiments” before this subsection, so we can refer to Fig.3G before Fig.4. Also in this new subsection, we summarized the statistical results that were originally only reported in Table S15. (See L174-183). Further, minor edits to clarify the manuscripts were made in the revised manuscript. Please see the track-changes version (tracked_changes.pdf) to confirm all changes.

Reviewer #1 (Remarks to the Author):

The study by Sugiyama, Schweighofer, and Izawa aims to establish that reinforcement learning works as a metacognitive process to affect motor learning. They formulated a hierarchical model that uses reinforcement to modulate learning and retention rate and confirmed with empirical evidence that monetary reward can affect people's motor adaptation during training and interleaved learning probe sessions. Furthermore, positive and negative rewards affect motor learning and retention differently,

consistent with previous work. To support the new model as a unified theory for motor learning, the authors also provide model simulations that qualitatively resemble the major findings of three previous motor learning research studies.

Overall, the paper is well written with a high-quality data presentation. The two experiments are beautifully designed to show that reinforcement learning indeed affects motor learning after the training period, critical evidence for a meta-learning effect. However, this effect is weak given that the probe session is interleaved with the training session, and the effect size is small (see below). The model is also reasonable.

However, I have two major concerns regarding the new contribution of the present work.

We thank the reviewer for the positive comments and insightful feedback. In the revised manuscript, we emphasized the connection between our model and previous motor learning models via new simulation results presented in the revised supplementary materials.

R-Q1)

The theoretical novelty about the effect of reward on up and down-regulate adaptation.

The present model assumes that reinforcement hierarchically affects the low-level adaptation by regulating its learning rate and retention. The authors pointed out that they provide "the first empirical evidence for the existence of a reinforcement learning mechanism for motor learning policy in the human brain." But the fact that reinforcement changes motor learning policy (e.g., reflected as learning rate or retention) is a well-taken idea in the field (e.g., work done by Galea and colleagues in recent years). Or, we can step back and say that the present study provides evidence that reinforcement learning works hierarchically as a meta-learning process for low-level learning processes (i.e., the processes based on sensory prediction error, SPE). But the hierarchical learning structure has also been proposed before. Kim and Ivry (2019 eLife paper, which the current paper should quote) showed that target error (TE) serves as a reinforcement signal to modulate the learning rate in an SPE-based learning task. They even proposed an alternative non-hierarchical structure for the effect of reinforcement on adaptation, i.e., TE-based reinforcement learning and SPE-based learning work in parallel to drive learning. Their data sets could not differentiate these two model alternatives.

I really like the author's emphasis on the "metacognitive process" in motor learning. In fact, besides reward signals shown here, other metacognitive processes also affect motor learning, including contextual inference, conceptual learning, or simply task instructions. These results reflect that motor learning is not just implicit, non-verbal, and procedural learning but also incorporates explicit, conscious, and declarative components. But the paper frames its contribution or novelty by oversimplification or neglect of previous work. For example, it states that the hypothetical goal of motor learning is error minimization (Line245). This is too much of a simplification of the field: starting from Krakauer et al. 2006 seminar paper on unintentional adaptive drift based on SPE, the area

already takes a multiplicative view of motor adaptation: it contains multiple components, including minimization of different errors even sacrificing task performance, reinforcement learning, use-dependent learning, etc. Similarly, when the authors claim that previous theories for learning speed regulations only consider properties of errors (Line236), the studies on reward or use-dependent learning are neglected. Thus, it is questionable whether the current study provides a conceptual step-up in the realm of motor learning as compared to similar studies, say, work done by Galea and Ivry groups in recent years.

R1-A1)

We thank the reviewer for pointing out this issue and providing a comprehensive set of references (McDougle-2015, Kim-2019, and Albert-2022, given by references 5, 13 and 6 in the revised manuscript). The omission of these critical papers from our reference list gave the reviewer the impression that we oversimplified the contributions of the previous motor learning literature. We apologize for this omission. We revised the manuscript to discuss these papers in addition to the new simulation results of these models (see also response R1-A2).

We also thank the reviewer for finding an interest in the meta-cognition of motor learning. Although the term meta-cognition is used broadly, we focus on the meta-cognitive function of monitoring and controlling a learning system, which was proposed in educational psychology. Our theorization and demonstration of this meta-cognition for implicit motor learning is novel and gives a conceptual advance in neuroscience and cognitive science. Specifically, we theorized this meta-cognitive system as a reinforcement learning of motor learning property. However, the reviewer questioned the "theoretical novelty" of our meta-cognition of motor learning.

To highlight the theoretical novelty of our study, we highlight two critical theoretical differences with previous studies that addressed reinforcement/punishments in motor learning (Galea-2015 and Kim-2019). The first critical theoretical difference is that our proposed system can be categorized as an instrumental learning system, whereas previous studies proposed Pavlovian learning systems. As nicely summarized in a review by Peter Dayan (reference 55), the human and animal reinforcement system are composed of Pavlovian and Instrumental learning systems. The Pavlovian system responds to valence in a hardwired manner. For example, in the Pavlovian system, the response gain is upregulated by punishment and downregulated by reward, and this relation is fixed. In contrast, the Instrumental learning system can learn to regulate the response gain to maximize rewards or minimize punishment by considering action-outcome structures for goal-directed tasks. For example, even for the same punishment feedback, if the higher response gain results in a larger punishment, the

system learns to reduce the gain, whereas if the higher gain results in a lower punishment, the system learns to increase the gain. Computationally, such an instrumental learning system has been modeled by Sutton-Barto's reinforcement learning algorithm. Our derivation of the reinforcement learning rule for controlling motor learning is novel in that a formulation of instrumental learning of the motor learning system has not been formalized previously. In contrast to our proposed model, the experimental result of Galea-2015 suggested a simple Pavlovian system, where the punishment upregulates the learning rate. Similarly, in Kim-2019 (Adaptation Modulation model, Figure 4b and Eqn.3 in Kim-2019), the relationship between punishment (miss the target) and the gain modulation (switching two between gains by hit or miss) is also fixed. This model is, therefore, also categorized as Pavlovian. (See L315-328).

The second (yet related) critical difference is that, in the proposed meta-cognition of motor learning, higher cognitive system monitors and controls lower adaptive system based on signals for memory update provided from the lower system (Fig. 1B). This requires a bidirectional tight coupling between higher and lower systems. In contrast, Kim-2019 suggests only unidirectional modulation where the miss of the target increases the learning gain. Importantly, our theory is derived from a novel assumption: "motor learning is a sequential decision-making process in the memory space regarding learning-outcome structure." This assumption has not been proposed previously in the motor learning literature (e.g., Izawa-2011, Galea-2015, Kim-2019, Albert-2022). This revised assumption of motor learning processes enabled us to reformulate motor adaptation subject to the optimization process. This led to our innovative theoretical conclusion that the error minimization principle holds only when motor learning is associated with maximizing rewards or minimizing punishments. Thus, our theory provides, for the first time, a computational level account of the modulation of motor learning according to David Marr's levels: higher system specifies the computational problem that the system is solving (maximizing rewards or minimizing punishments), and the lower level uses motor learning to solve that problem. (See L256-264)

Thanks to these advances, our model can provide unifying account for the various motor learning profiles reported in the previous results which the recent models suggested by the reviewer cannot. Specifically, since Mazzoni 2006 (reference 4) proposed an effect of the cognitive strategy on motor adaptation, it has been assumed that the strategy component and implicit component are independent, generating motor commands additively (McDougale-2015, Kim-2019, Albert-2022) or multiplicatively (Kim-2019). In the revision, we simulated the models proposed McDougale-2015, Kim-2019, and Albert-2022 to account for the variable learning profiles reported in Herzfeld-2014, Leow-2020, and Galea-2015 (see new **Figs. S5-6**). Unlike our model, these previous models with explicit strategy and implicit learning could not explain these previous results. Thus, our new concept of meta-cognitive mechanism for motor learning based on the reinforcement learning of motor learning is highly distinct from the additive or

multiplicative cognitive strategy effects. We revised the manuscript to discuss these points in the discussion section (L303-314). Please see also R1-A2.

As a final note, we also controlled the use-dependent factor (reference 64) by randomizing the target position. The original manuscript contained a citation related to use-dependent learning (reference 54 in the original manuscript).

R1-Q2)

Whether the presented model can provide a unified theory to account for various previous results should also be questioned.

The authors showed model simulations that qualitatively resemble the findings in three previous studies. For the first two studies (Herzfeld-2014 and Leow-2020), the current study assumes that task error serves as negative reinforcement feedback. This treatment should be cautioned, though. Target error has been proposed to drive explicit strategic learning (McDougle-2015) and implicit learning (Albert-2022). Indeed, the current paper tried to avoid strategic learning by using small perturbations (5 and 7 degrees rotation) and to focus on SPE-based implicit learning. However, having TE as a reinforcement signal is a conceptual change for the field. Thus, the authors should provide adequate evidence to support this new conceptualization of TE. The evidence here is still qualitative, based on model simulations for three previous data sets. First, there is no testing of alternative models, e.g., the Albert-2022 model; there is no direct evidence based on the new experiment (see below). Second, calling TE as negative reinforcement does not differ from the explanation offered by the original studies. For example, the first modeled study here (Herzfeld-2014) has its own modeling, which in spirit is the same as the meta-learning model: the consistency of perturbation affects adaptation performance (TE); thus, the learning rate is changed to minimize error. The reinforcement model here asserts almost the same but using different wording: the task error serves as punishment which modulates the learning rate when the consistency of perturbation changes. A similar reconceptualization of TE is used in simulating the results of Leow-2020. To convincingly "re-formulate" TE as punishment instead of as a straightforward performance error to be minimized, the authors should provide direct evidence.

Note the two new experiments presented here did not model TE (in fact, it is clamped to zero) as reinforcement but rather used monetary reward to emulate the reinforcement effect. This design cleverly dissociates the sensory prediction error and the reward in adjacent trials to show the meta-learning effect; the natural consequence is that it cannot serve as evidence for the role of TE as a reinforcement. Given that the role of TE is hotly debated with multiple models available, we shall be cautioned when the new theorization is only supported by qualitative analysis. If the meta-learning model is indeed a persimmons model, we would expect to see at least some kind of model comparison to show its superiority over other competing models.

R1-A2)

We thank the reviewer for raising this important issue. The reviewer pointed out that the concept of task error (TE) as a driving input to motor memory has been established recently (McDoughle 2015, Albert 2022). Our aim of the present paper was not to question this established idea. Instead, we argue that TE includes two different pieces of information: driving input and negative reinforcer (i.e., punishment). Specifically, we propose that the continuous negative reinforcer contained in TE provides a unified account for the three very different motor learning profiles seen in Herzfeld-2014, Leow-2020, and Galea-2015.

Specifically, we presented new simulation results (shown in **Fig. S5-6** in the revised supplementary materials) to explain the data in Herzfeld-2014, Leow-2020, and Galea-2015 with the recent computational models suggested by the reviewer, which only contain a TE-driving effect (**Fig. S5**, McDoughle-2015, Kim-2019, Albert-2022). Note that the free parameters were manually tuned in the original simulation (**Fig. 4**) because its purpose was qualitative replication. For the new simulation, we performed parameter optimization to compare the previous models with the proposed one not only qualitatively but also quantitatively. In these new simulations, we demonstrated that our meta-learning model outperformed these computational models with the driving input factor (**Figs. S5-6**), both when assessed qualitatively (**Table S17**) and quantitatively via the root mean squared error (**Table S18**). This model comparison demonstrated that the meta-learning model outperforms the previous models in abilities to explaining various motor learning profiles.

In addition, in the revision, we also simulated a new “hybrid” model in which our meta-learning model is combined with the previous model where TE drives explicit strategy. The simulated learning profiles with this hybrid model replicated Herzfeld-2014, Leow-2020, and Galea-2015 as well as the original meta-learning model, suggesting that the punishment factor and the driving input factor could coexist in motor learning (**Fig. S6E**). This supports the idea that TE is involved both as the negative reinforcer (punishment) and as the driving input to motor learning.

Fig. R1 below details the simulation results of the hybrid model. In this model, the sensory prediction error updates implicit memory which is modulated by TE dependent meta-learning model. In addition, TE drives explicit memory. Then, the sum of these two memories generates the motor output. There were significant contributions of implicit memory (meta-learned) in the replication of Herzfeld-2014, Galea-2015, and Leow-2020. We revised the manuscript by adding this point to discussion (L303-314).

Figure R1.

Total memory that generates observed movements (faded) consists of implicit memory (dashed) and explicit memory (dotted) in the extended meta-learning + target error model. Contributions of implicit memory and explicit memory are determined through optimization of free parameters. See Supplementary Text for detail of the simulation. A. Herzfeld-2014. B. Leow-2020. C. Galea-2015. Lines indicate means. For visualization, standard errors are omitted.

Although our meta-learning model and hybrid-model outperform the previous models driven by TE, one criticism is that there is no direct evidence that TE contains the negative reinforcer. However, experimentally dissociating this punishment factor from the driving input factor, which is inseparable in the TE feedback information, is impossible. Thus, our approach was to provide punishment as monetary feedback by masking visual TE. Another possible way to examine the punishment hypothesis would be to manipulate TE directly by perturbing it independently from SPE and to estimate the contribution of these two factors contained in TE via model fitting methods. Indeed, Leow-2020 manipulated TE effectively by randomizing and erasing it during a typical motor adaptation task; however, the punishment factor was not discussed in their original manuscript. In our new model comparisons (**Figs. S5-6**), we demonstrated that the meta-learning system with the hypothesis of punishment factor in TE was necessary to replicate Leow-2020. In contrast, motor learning driving by TE (Mcdougale-2015, Kim-2019, Albert-2022) cannot replicate the Leow-2020 results (**Figs. S5-6**).

Specifically, in Leow-2020, there were three conditions with different types of TE manipulation. In the StdTE condition, because TE was presented without any manipulation, TE was approximately equal to a difference between 30° rotation and memory measured as reach direction (i.e., 30° – memory). Therefore, within a first few cycles of adaptation training, TE decreased to <20° as memory increased to >10°, and then TE kept decreasing to 0° as memory kept increasing to 30° (See **Fig. 4C** or their original figure). In this condition, learning is associated with the decrease of TE. In contrast, in the RandomTE condition, TE was randomly presented in the range of 20~30° independent of reach direction. Therefore, TE was larger in RandomTE than in StdTE, except in first few cycles of training. Importantly, in RandomTE, TE was presented independent of learning. Finally, in the NoTE condition, TE was always set at 0° via the target jump technique. Therefore, TE was smaller than in StdTE and RandomTE.

Remarkably, this order of TE size (NoTE<StdTE<RandomTE) was not consistent with adaptation amplitude (NoTE=RandomTE<StdTE). Namely, Leow-2020 reported that adaptation in RandomTE was smaller than StdTE, and adaptation was comparable between RandomTE and NoTE (**Fig. 4C**). These contradict the predictions made by TE-driven models (NoTE<StdTE<RandomTE). Namely, because the average TE is the largest in RandomTE and the smallest in NoTE, TE-driven models incorrectly predict the largest overall memory in RandomTE and smallest memory in NoTE (**Fig. S5B-C**). In contrast, because learning-punishment structure was corrupted in RandomTE, our meta-learning model replicated no-upregulation of learning in this condition (**Fig. S6D**). Similarly, because TE was masked in NoTE, our meta-learning model replicated no-upregulation of learning also in condition (**Fig. S6D**). In other words, since learning-punishment structure was preserved and TE was

available only in StdTE, our meta-learning model replicated upregulation only in this condition. Thus, Leow-2020 provides empirical evidence of our hypothesis that TE contains a punishment factor.

Finally, this hypothesis of punishment factor is supported by previous observations of neural activities. First, Diedrichsen et al. (reference 23) reported that TE activated a part of the striatum, while SPE activated the cerebellum and the posterior parietal cortex. Thus, TE information is processed in the basal ganglia, which is considered responsible for reinforcement learning. Second, when TE is explicitly noticeable (McDougle-2015), the noticeable error in goal-directed behavior is aversive (reference 34), which may decrease the dopamine in the ventral tegmental area (reference 32 and reference 33). Thus, TE containing the negative reinforcer factor is likely represented in the basal ganglia and processed by reinforcement learning system (L292-302).

Minor:

R1-Q3)

1) The effect of meta-learning is mainly reflected in initial update and retention during the Probe task (Figure 2E). The initial update is only a point estimate of the learning rate since it is merely the response amplitude to the first perturbation. Another concern is that the retention is an aggregation of the memory decay process as it is the average of hand direction during the whole error-clamp session. This retention (Figure 2E) is not comparable to the retention rate (Figure 2F) since the former reflects both the learning extent (before decay) and the decay rate while the latter is the rate itself.

R1-A3)

We thank the reviewer for pointing out these issues. The reviewer's interpretation of these indexes (Initial update, Retention learning rate, and Retention rate) is correct. Mainly, "Retention" includes both the learning effect and the forgetting effect. Thus, this is not comparable with the estimated retention rate (α). This is also the reason why we conducted the model-based estimation of α . To avoid confusion, we revised the manuscript by changing Fig 2E's indices name for the probe task phase. We use "Error clamp trials" instead of "Retention" in the revised manuscript and explicitly note that this value includes accumulating the memory update and the forgetting. We also explicitly noted that the initial update is the point estimate of the learning rate, which reflects only the initial response to the error (L152-155).

R1-Q4)

2) The use of 7-degree perturbation as a test of generalization is a nice design, but it is not sufficient to rule out the possibility that the reinforcement learning system recognizes the rotation (5 degrees) as the

one experienced before. This is because 5 and 7 degrees are similar in amplitude. This is reflected in Figure S3 where the error sensitivity change covers a wide range of error sizes, with the 7-degree rotation in the envelope. The more convincing generalization test for a meta-cognitive model should use a rotation in the opposite direction.

R1-A4)

We thank the reviewer for pointing out this issue. As the reviewer pointed out, the comparison with 5 degrees rotation and 7 degrees rotation is not sufficient to thoroughly test the perfect generalization ability over the error space. However, the objective of evaluating the generalization ability is to show a partial generalization ability to similar sizes of error, which has been seen in human motor learning properties (reference 53, 54, and 8). As the reviewer pointed out, **Fig. S3** demonstrated a local generalization function between and beyond 5- and 7-degrees rotations. This analysis, based on the previous literature (references 53, 54, and 8) is a convincing result of generalization function over error inputs: it suggests that meta-learning involves a neural policy function based on the neural basis functions with receptive fields (Equation 2 and the equation on L562), ruling out the possibility of the simple cue-response map. We have added text to discussion (see new text in L284-291).

R1-Q5)

3) The primary task goal for the participant is to maximize reward (instantiated by the R trials). According to the authors, the interleaved S trial induced implicit learning since participants did not report a re-aiming strategy in the post-task written questionnaire. However, the authors also stated that "...participants were informed about stimuli and feedback in the Null, S, and R trials and practiced each type of trial before the task.". Thus, the participants knew rotated feedback would be given in S trials, i.e., they knew conceptually that their actions would be affected by rotations or movement direction changes. Thus, I wonder whether a reward-modulated recall of motor solutions partially causes the meta-learning effect. This possibility is corroborated by the abnormally fast learning in the Probe phase: the adaptation to 7-degree rotation is almost finished in the first trial (shoot up to 5 degrees), which is drastically different from typical error-clamp learning.

Note the additional target jump would only reduce the learning instead of accelerating it.

R1-A5)

We thank the reviewer for pointing out these issues. The participants were instructed about the presentation/omission of the cursor and the score feedback, but they were not informed about the manipulation of the cursor, including error clamp and rotation. Therefore, they did not know that the relationship between their action and rotations. This sentence was revised to make this clearer (L631-636).

We do not believe that a reward-modulated recall of motor solutions partially causes the meta-learning effect for the following reason. We first note that, whether or not there exists memory that is modulated by reward, a cue signal (TE) is necessary to recall memory (reference 15). However, in our task, Promote and Suppress (originally called Lrn and NLrn. See suggestion for renaming in R2-Q6) showed significantly different initial updates both in Training and Probe despite the lack of explicit cue signal (TE or score) in the initial exposure to perturbation. Thus, the modulated initial response to the pure sensory prediction error was achieved only by updating the learning rate but not by memory recall. This is further supported by our error-sensitivity (generalization) analysis which showed that the meta-learning was performed by the gain tuning of the neural policy function, which affects the change of the generalization function of the motor learning effect (**Fig. S3**).

The reviewer noted the “abnormally fast learning in the Probe phase” may have been due in part to our (poor) figures: we did not mark each trial data in the original version of **Figs. 2D, 3D** (this part has been revised), and the response to the 1st and 2nd perturbations occasionally appear on the same slope. We have now revised the figures with marking on each data point. As shown in revised **Figs. 2E, 3E**, the size of the 1st response in the 1st block was approximately 40% of the perturbation (3 degrees), which is comparable with previous results (reference 51).

Finally, we also note recent evidence showing that implicit learning is much faster than traditionally thought, reaching an asymptote in a few trials (reference 52), which is consistent with our data that the initial response of our data with 40% of the perturbation. This further suggests that the initial response in our task was mediated by implicit motor adaptation. This initial response was modulated gradually over blocks (**Figs. 2E, 3E**) and reached 5 degrees in the last block in Promote (Lrn) condition. Since this initial update is generated from the policy function of sensory error, not to the explicit cue signal (TE or reward), the initial update modulation was due to the update of the gain tuning of the policy function. We revised the manuscript to state this in the discussion (L268-277).

R1-Q6)

Line 577: 10 and 30 null trials before the training cycle and Probe? The ten null trials are not specified in the task design.

R1-A6) It should have been 10 Null trials **in** each training cycle. Thank you for pointing out the mistake. We have corrected it (L667).

R1-Q7)

The group and block effects were estimated by Bayesian inference with some modeling assumptions. What would the result be like if we used conventional statistical methods like ANOVA? Why not use

the first Probe before training as a baseline? Or even use it to estimate $\alpha_{[g,base]}$ and $\beta_{[g,base]}$?

R1-A7)

Since our data does not satisfy the normality assumption, we could not use ANOVA. In addition, simple ANOVA does not dissociate the individual variability from the group variability. Behavioral scientists have started to caution against using the two-step analysis where the parameter, such as learning rate, is estimated in the 1st step and then conducted the group-level statistical analysis in the 2nd step ignoring the variance in the 1st step, which often biases the estimate of the effect. Instead, they have recommended using Hierarchical Bayesian modeling (reference 67). Using Bayesian modeling, we can incorporate the motor noise, the state update noise, and the across-participants variability in a unified and systematic manner (Equation 6, L717). For these reasons, we selected the Bayesian framework and hope this will encourage scientists in this field to shift to using the Bayesian method from using two-step analysis. Importantly, we confirmed that the results did not depend on the prior probabilities. This application of the Bayesian method to motor learning was also successful in our previously published paper (reference 30) by the last author as well as in a recently uploaded report in MedRxiv 2022 by the second author.

Also note that, as shown in **Fig. 1F**, the Probe before the training without the score feedback, i.e., baseline, was already included in this model-based analysis.

R1-Q8) Line652: what kind of model is used if linearity is not assumed?

R1-A8) We thank the reviewer for the question. With the linearity assumption, $\alpha_{[g,block]}$ and $\beta_{[g,block]}$ are calculated from $\alpha_{[g,base]}$, $\beta_{[g,base]}$, γ_g^α , and γ_g^β (Eqn 7). Without the linearity assumption, instead of estimating γ_g^α and γ_g^β , $\alpha_{[g,block]}$ and $\beta_{[g,base]}$ are estimated directly, just like $\alpha_{[g,base]}$ and $\alpha_{[g,base]}$. The left panels of **Figs. 2F** and **3F** show the results of this estimation without the linearity assumption. We have edited the methods and explicitly stated this in L744-745.

R1-Q9) Line669: trial-trial series?

R1-A9) trial-to-trial series. Thank you very much.

R1-Q10) Line 687: trails

R1-A10) corrected. Thank you very much.

R1-Q11) Line735: it is unclear what this equation means and how the errors are converted from the Herzfeld-2014 study.

R1-A11) To simulate Herzfeld-2014, the prediction error in the memory space should be transformed into a visual error. Since this is the force field adaptation task, we transformed it

by taking the biomechanical nature of the arm into consideration following (reference 53). We revised these equations to make this part clearer (L824-838).

R1-Q12) Figure 1F: The first block is marked with a different color compared to the later three blocks; this is confusing. Is the first block identical to other blocks except that it has a Probe phase before?

R1-Q12) The First block is for the baseline condition where no monetary feedback was provided (indicated with "No reward"). Thus, we thought it was reasonable to use a different color. This was added to the figure legend.

Reviewer #2 (Remarks to the Author):

Sugiyama / Izawa review

Sugiyama and colleagues propose a theory of meta-learning in the context of motor adaptation. Specifically, they propose a principled computational model of how the learning rate and retention of motor adaptation might be altered through experience. Their model nicely accounts for previous experimental findings relating to modulation of learning, and is confirmed in two new experiments that were specifically designed to test the predictions of the model by increasing reward/punishment feedback on trials in which participants had altered their behavior by a greater amount from the preceding trial.

This is an innovative perspective on motor learning. The paper is very clearly written. The experiments are cleverly designed and well conducted. The experimental results are clear. The theory also convincingly explains a number of well-known results relating to changes of learning rate. The discussion is thoughtful and insightful. One weakness of the paper is the quite small group sizes (8 participants), but the key contrast between groups is replicated in Experiment 2. Overall, this is an impressive and insightful paper.

I have a few (relatively minor) concerns and suggestions:

We thank the reviewer for the supportive comments. We recruited 80 participants in the experiments (40 healthy adults in experiment 1 and another 40 in experiment 2) , which allowed us to demonstrate reliable statistical tests.

R2-Q1)

A weakness is that the figures show only group-level data. There are no figures that indicate individual participant performance, so it is difficult to assess whether the results were driven by a minority of participants that had strong effects or whether all participants behaved similarly. I think the findings are still interesting either way, but it would be informative to be able to see some individual-level data, even if this is in the supplemental materials.

R2-A1)

We thank the reviewer for pointing out this issue. We added new plots for the individual data in **Fig. S4**, corresponding to the reach directions and score performance in **Figs. 2B, 2E, 3B, 3E, Figs. 2C and 3C**.

R2-Q2)

It's well-established that motor learning comprises both implicit and explicit components, but the narrative and discussion are not totally clear about which component the theory and experiment are seeking to describe.

R2-A2)

We thank the reviewer for pointing out this issue. This was also pointed out by Reviewer 1. We, therefore, refer the reviewer to R1-A1,A2. We notably added text in the discussion section to discuss how our model differs from the previous motor learning models that comprise both implicit and explicit components (L303-314).

R2-Q3)

Some previous work has suggested that certain types of meta-learning only occur when participants become explicitly aware of the experimental manipulation (e.g., Manley et al., 2014). I do wonder whether participants became explicitly aware or not in this case. It doesn't seem that participants were asked about awareness of the manipulation, and in any case, such insight should not have influenced the probes.

R2-A3)

Indeed, human reinforcement learning can be achieved via an implicit learning process in random environments where the participants were instructed to increase scores without explicit instruction of the action-outcome structure (references 49 and 50, which were added in the revised version in response to the reviewer). Because we did not explicitly instruct the learning-outcome structure (Promote(Lrn)/Suppress(NLrn)) to the participants and because the actual trial-to-trial sequence of the score was partly random due to the natural variability of motor movements, it was hard for the participants to recognize the task structure explicitly. Because the learning-outcome structure was simple and presented in a low dimensional space, the participants would have been able to solve this problem easily if they had recognized this

structure explicitly. Nevertheless, all participants exhibited imperfect score performance and gradual increase of it (Fig. S4). This supports that they were not aware of the task structure and did not solve this problem explicitly. Rather, the increase of the performance was mediated by the gradual update of learning policy, i.e., meta-learning. Although how explicit instruction influences meta-learning is an interesting question, we have not examined this in our task. (L271-274)

R2-Q4)

The introduction raises the apparent paradox that humans can cognitively modulate the learning rate of an implicit process. I don't really see this as a paradox unless the authors are proposing that the meta-learning is explicit but the adaptation itself is implicit. If this is implicit modulation of implicit learning, or explicit modulation of explicit learning, then I don't see any puzzle.

R2-A4)

We thank the reviewer for raising this point. According to the conventional study of meta-cognition, where students in the classroom were encouraged to recognize their “explicit learning ability” to promote its ability explicitly, meta-cognition of implicit motor learning sounds paradoxical. Here, we aim to highlight that the meta-cognition of motor learning has not been examined yet. We revised this sentence to make this point clear (L37-41).

R2-Q5)

line 64-71 - I would suggest being more explicit about alpha and beta here - i.e., include an update equation. At the moment, if the reader has a reasonable understanding of conventional motor learning models their meaning will be obvious, but if not, then it will require consulting the methods. It's also a little odd to define update equations for alpha and beta when they haven't yet been concretely defined.

R2-A5)

We thank the reviewer for this suggestion. We revised the manuscript by adding the equations to define alpha and beta (L66-69).

R2-Q6)

"Learn" and "No-learn" group names are slightly inaccurate and potentially confusing. Both groups learn, after all. I worry some readers might become confused by this. So the authors might consider alternative names for these groups. e.g. "Promote learning", "Suppress learning."

R2-A6)

We thank the reviewer for this suggestion. We revised the paper to use "Promote" and "Suppress" to call these two groups.

R2-Q7)

The reason "S" and "R" trials are called as they are "Sensory Error" and "Reward" is only made clear in the methods. It would be helpful to briefly mention this in the results also to make it easier to remember what they are. Also, I wonder if calling "S" trials "E" trials instead might be more intuitive, since the 'Sensory' part is a bit superfluous - but this is entirely a suggestion.

R2-A7)

Thank you very much for this suggestion. We used E instead of S and then clearly stated their definition in the result section in the revised manuscript.

R2-Q8)

"reinforcement learning of motor learning" is a slightly awkward phrasing. Perhaps something like "reinforcement learning of motor learning properties/parameters", at least at some points in the text.

R2-A8)

Thank you very much for this suggestion. We used "reinforcement learning of motor learning properties" throughout the manuscript (L23, L138-139, L499, L877).

REVIEWERS' COMMENTS

Reviewer #1 (Remarks to the Author):

The revision has addressed all my previous concerns except some minor problems. The manuscript is a joy to read now. The newly added motor comparison results complete the main ideas the paper tries to convey, though I could not assess them in detail since the codes are unavailable. I only have some minor suggestions below.

Minor :

The reward function $r()$ is not specified for the model fitting of others' datasets (Figure 4). I guess that the reward function is assumed to be acquired to the participant for Exp1 and 2, say, by trial and error. Then, is it assumed that participants know the outcome (TE) for changing x for datasets in Figure 4? Note that "knowing" the exact form of a reward function is equivalent to mastering the task. Of course, we can argue that because the reaching adaptation here is straightforward, we thus can simply assume it is known to the system. Will the author make the model codes available?

The proposition that "motor learning is a sequential decision-making process in the memory space regarding learning-outcome structure" should be related to the paper by Konrad Kording, titled "decision theory: what "should" the nervous system do?" That 2007 review paper tries to link motor control, not just motor learning, to decision-making.

Line 41: "How can we monitor and control motor learning that we are not aware of?" The wording needs to be modified here. Motor learning is not something we are typically not aware of. It is hard to find cases that are completely unaware to the learner. Unawareness is not the same as implicitness. I guess here it refers to perturbations that are unaware.

Line42: retentions?

Line185 : "unification of previous results..." sounds strange. Unified explanations for previous results?

Line186: "reinforcement learning mediated meta-learning of motor learning"? —> reinforcement learning or feedback affected the properties of motor learning?

Line302: a comma missing

Line302: a comma missing

Reviewer #2 (Remarks to the Author):

The authors have addressed all my previous concerns and improved the paper.

A few very minor things:

line 42 - "retaintions" -> "retention"

line 66 - "accelerated/decelerated motor learning may be a result from valuing/devaluing it" - this part of this sentence didn't quite make sense to me. Maybe reword to "accelerated/decelerated motor learning may RESULT IN valuing/devaluing it."?

line 67 - "Here, we sought A minimal framework"

Point-to-Point Response to the Reviewers

Manuscript ID: NCOMMS-22-36133-A

“Reinforcement learning establishes a minimal metacognitive process to monitor and control motor learning performance.”

by Taisei Sugiyama, Nicolas Schweighofer, Jun Izawa

We sincerely appreciate the editor and the reviewers' further comments and concerns on our manuscript. We have revised and improved the manuscript as suggested. In the following, we provide a point-to-point response to each question raised.

Reviewer #1 (Remarks to the Author):

The revision has addressed all my previous concerns except some minor problems. The manuscript is a joy to read now. The newly added motor comparison results complete the main ideas the paper tries to convey, though I could not assess them in detail since the codes are unavailable. I only have some minor suggestions below.

We appreciate the reviewer's comment. We discovered that, although we had uploaded the simulation scripts on Figshare prior to submitting our first revised manuscript, the hyperlink was incorrectly set, leading to an error page. We sincerely apologize for the mistake. We have corrected the hyperlink in the revised manuscript.

Minor :

R-Q1)

The reward function $r()$ is not specified for the model fitting of others' datasets (Figure 4). I guess that the reward function is assumed to be acquired to the participant for Exp1 and 2, say, by trial and error. Then, is it assumed that participants know the outcome (TE) for changing x for datasets in Figure 4? Note that “knowing” the exact form of a reward function is equivalent to mastering the task. Of course, we can argue that because the reaching adaptation here is straightforward, we thus can simply assume it is known to the system. Will the author make the model codes available?

We appreciate the reviewer's comment. Our meta-learning algorithm is based on a model-free Reinforcement Learning (RL) framework, which does not require an explicit model of the environment or reward functions. Instead, it optimizes behaviors, namely, memory updates, merely through a process of trial and error.

In Figure 4, the simulation was conducted using the same meta-learning agent, which does not incorporate a function of TE. Its sole reliance is on repeated trial and error to minimize TE. This approach is consistent with the main simulation in Figure 1, which is predicated on the assumption that the participant was unaware of the changes in the outcome resulting from modifications to 'x'.

Therefore, it would be incorrect to infer from this model that the participants must be aware of or learn the exact form of the reward functions or TE as a function of memory. We have now explicitly stated this in the methodology section of our paper.

Please note that the model codes are accessible online via Figshare, as previously indicated.

(L414-416, L710-711)

R1-Q2)

The proposition that “motor learning is a sequential decision-making process in the memory space regarding learning-outcome structure” should be related to the paper by Konrad Kording, titled “decision theory: what “should” the nervous system do?” That 2007 review paper tries to link motor control, not just motor learning, to decision-making.

We thank the reviewer for referring to the paper. We added this reference in line 275. The theory discussed in Kording 2007 is indeed about decision-making in motor control, which differs from our theory about decision-making in memory updating. Although these two use the same term “decision-making” for motor behavior and seem parallel, they are conceptually different. Our theory is novel since the idea of decision making of memory update is has not be proposed previously. We have added a short paragraph to the discussion to clarify this.

(L274-278)

R1-Q3)

Line 41: “How can we monitor and control motor learning that we are not aware of?” The wording needs to be modified here. Motor learning is not something we are typically not aware of. It is hard to find cases that are completely unaware to the learner. Unawareness is not the same as implicitness. I guess here it refers to perturbations that are unaware.

We thank the reviewer for the comment. It indeed refers to unawareness to perturbation, which implies implicit motor learning. We have modified the expression, stating, “How can we monitor and control implicit motor learning?” (L42)

R1-Q4)

Line42: retaintions?

Corrected. (L44)

R1-Q5)

Line185 : “unification of previous results...” sounds strange. Unified explanations for previous results?

We thank the reviewer for pointing out the problem. We have changed the part according to the suggestion. (L192)

R1-Q6)

Line186: “reinforcement learning mediated meta-learning of motor learning”? — > reinforcement learning or feedback affected the properties of motor learning?

We thank the reviewer for the suggestion. We now state, “... reinforcement learning affected the properties of motor learning ...”. (L193-L194)

R1-Q7)

Line302: a comma missing

Corrected. (L318)

Reviewer #2 (Remarks to the Author):

The authors have addressed all my previous concerns and improved the paper.

A few very minor things:

R2-Q1)

line 42 - "retaintions" -> "retention"

Corrected. (L44)

R2-Q2)

line 66 - "accelerated/decelerated motor learning may be a result from valuing/devaluing it" - this part of this sentence didn't quite make sense to me. Maybe reword to "accelerated/decelerated motor learning may RESULT IN valuing/devaluing it."?

We thank the reviewer for the comment. By that we meant that humans valued/devalued motor

learning, which would promote/suppress motor learning and influence its learning speed. We have modified the sentence to clarify this point. (L54)

R2-Q3)

line 67 - "Here, we sought A minimal framework"

Corrected. (L56)